# Geometry-Informed Neural Networks

Arturs Berzins [* 1 2]  Andreas Radler [* 1]  Eric Volkmann [* 1]
Sebastian Sanokowski [1]  Sepp Hochreiter [1]  Johannes Brandstetter [1 3]

## Abstract

Geometry is a ubiquitous tool in computer graphics, design, and engineering. However, the lack of large shape datasets limits the application of state-of-the-art supervised learning methods and motivates the exploration of alternative learning strategies. To this end, we introduce geometry-informed neural networks (GINNs) – a framework for training shape-generative neural fields *without data* by leveraging user-specified design requirements in the form of objectives and constraints. By adding *diversity* as an explicit constraint, GINNs avoid mode-collapse and can generate multiple diverse solutions, often required in geometry tasks. Experimentally, we apply GINNs to several problems spanning physics, geometry, and engineering design, showing control over geometrical and topological properties, such as surface smoothness or the number of holes. These results demonstrate the potential of training shape-generative models without data, paving the way for new generative design approaches without large datasets.

## 1. Introduction

Recent advances in deep learning have revolutionized fields with abundant data, such as computer vision and natural language processing. However, the scarcity of large datasets in many other domains, including 3D computer graphics, design, engineering, and physics, restricts the use of advanced supervised learning techniques, necessitating the exploration of alternative learning strategies.

Fortunately, these disciplines are often equipped with formal problem descriptions, such as objectives and constraints. Previous works for PDEs (Raissi et al., 2019), molecular science (Noé et al., 2019), and combinatorial optimization (Bengio et al., 2021) demonstrate these can suffice to train models even in the absence of any data. The success of these data-free approaches motivates an analogous attempt in geometry, raising the question: *Is it possible to train a shape-generative model on objectives and constraints alone, without relying on any data?*

We address this question by introducing *geometry-informed neural networks* or *GINNs*. GINNs are trained to satisfy specified design requirements and to produce feasible shapes without any training samples. A GINN solves a structural optimization problem using *neural fields*, which offer detailed, smooth, and topologically flexible geometry representations, while being compact to store. This setup is analogous to physics-informed neural networks but with a high number of varied constraints: differential, integral, geometrical, and topological.

In contrast to both physics-informed neural networks and classical structural optimization, GINNs allow to generate multiple solutions by enforcing solution *diversity* as an explicit constraint. This is of high interest when applied to problems with solution multiplicity, e.g., induced by under-determinedness or near-optimality common in geometry problems. Connecting back to our research question, the proposed framework allows us to train neural fields that satisfy user-specified design constraints, and by adding diversity as an explicit constraint, we can generate a multiplicity of solutions. GINNs can thus be used as shape-generative models trained purely on constraints and without data.

We take several steps to demonstrate GINNs experimentally. We formulate a tractable learning problem using constrained optimization and by converting constraints into differentiable losses. We demonstrate the generality of the framework by solving several tasks, including an under-determined PDE, geometric optimization, and engineering design (Figures 4, 5). Figure 1 illustrates the task of designing a jet-engine lifting bracket, or geometrically – connecting cylindrical interfaces within the given design region. We show different GINNs trained with various smoothness and topology requirements, illustrating the robustness of the

---

*Equal contribution  [1]LIT AI Lab, Institute for Machine Learning, JKU Linz, Austria  [2]Part of this work was done while at SINTEF and Department of Mathematics, University of Oslo, Norway  [3]Emmi AI GmbH, Linz, Austria. Correspondence to: Arturs Berzins <berzins@ml.jku.at>.

*Proceedings of the 42$^{nd}$ International Conference on Machine Learning*, Vancouver, Canada. PMLR 267, 2025. Copyright 2025 by the author(s).

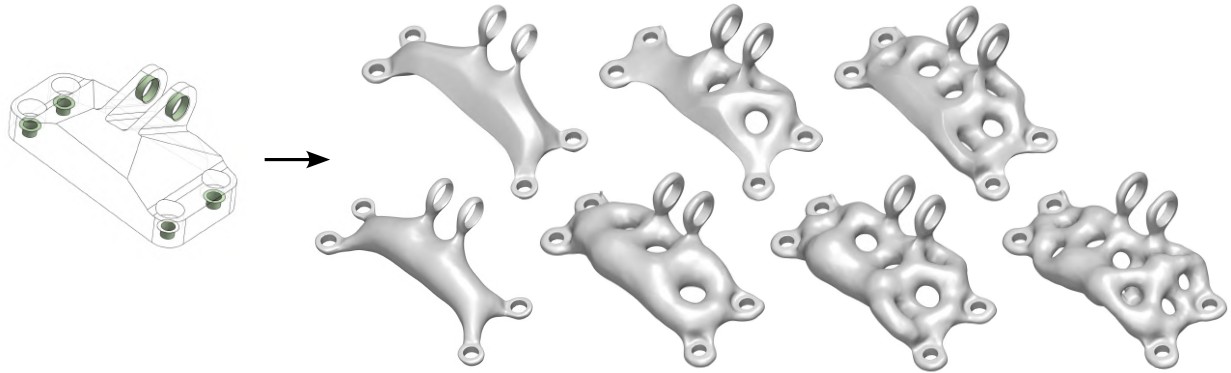

Figure 1: We train geometry-informed neural networks to *produce shapes satisfying geometric design requirements*. For example, we generate parts that connect the cylindrical interfaces within the sketched design region depicted on the left. To highlight the *user's control over the problem* and the solutions, we specify different additional requirements on the number of holes and surface smoothness. By complementing the design requirements with a diversity constraint, we can train a shape-generative model *without data* as further illustrated in Figures 3, 5 and 6.

constrained optimization approach and the user's control over the framework. Figure 6 shows a GINN trained on the same task but with an additional diversity constraint. Surprisingly, this constraint induces a structured latent space, with generalization capacity and interpretable directions.

We show that training shape-generative networks using constraints and objectives without data is a feasible learning strategy, paving the way for new generative design approaches without large datasets. Our main contributions are summarized as follows[1]:

1. We introduce GINN - a framework for training shape-generative neural fields without data by leveraging design constraints and avoiding mode-collapse using a diversity constraint.

2. We apply GINNs to several problems spanning physics, geometric optimization, and engineering design, investigating the user's control, the robustness of the method, key framework choices, and emerging latent space structure.

## 2. Related Work

We begin by reviewing and relating three important facets of GINNs: theory-informed learning, neural fields, and generative modeling.

### 2.1. Theory-Informed Learning

Theory-informed learning has introduced a paradigm shift in scientific discovery by using scientific knowledge to remove physically inconsistent solutions and reducing the variance

of a model (Karpatne et al., 2017). Such knowledge can be included in the model via equations, logic rules, or human feedback (Dash et al., 2022; Muralidhar et al., 2018; Von Rueden et al., 2021). Geometric deep learning (Bronstein et al., 2021) introduces a principled way to characterize problems based on symmetry and scale separation principles, e.g. group equivariances or physical conservation laws.

Notably, most works operate in the typical deep learning regime, i.e., with an abundance of data. However, in theory-informed learning, training on data can be replaced by training with objectives and constraints. More formally, one searches for a solution $f$ minimizing the objective $o(f)$ s.t. $f \in \mathcal{K}$, where $\mathcal{K}$ defines the feasible set in which the constraints are satisfied. For example, in Boltzmann generators (Noé et al., 2019), $f$ is a probability function parameterized by a neural network to approximate an intractable target distribution. Another example is combinatorial optimization where $f \in \{0, 1\}^N$ is often sampled from a probabilistic neural network (Bello et al., 2016; Bengio et al., 2021; Sanokowski et al., 2024).

**Physics-informed neural networks** (PINNs) (Raissi et al., 2019) are a prominent example of neural optimization. In PINNs, $f$ is a function that must minimize the violation $o$ of a partial differential equation (PDE), the initial and boundary conditions, and, optionally, some measurement data. Since PINNs can incorporate noisy data and are mesh-free, they hold the potential to overcome the limitations of classical mesh-based solvers for high-dimensional, parametric, and inverse problems. This has motivated the study of the PINN architectures, losses, training, initialization, and sampling schemes (Wang et al., 2023). We further refer to the survey by Karniadakis et al. (2021).

Same as PINNs, GINNs use neural fields to represent the solution. Consequentially, we also observe that some best

---

[1]Code is available at https://github.com/ml-jku/GINNs-Geometry-informed-Neural-Networks

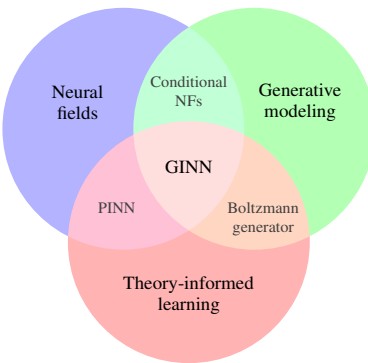

Figure 2: GINNs build on neural fields, generative modeling, and theory-informed learning.

practices of training PINNs (Wang et al., 2023) transfer to training GINNs. However, PINNs may suffer from ill numerical properties due to minimizing the squared residual of the strong-form different to classical PDE solvers (Rathore et al., 2024; Ryck et al., 2024). In contrast, GINNs share the same underlying formulation and numerical properties as classical topology optimization methods. In addition to a high number of various constraints (differential, integral, geometrical, and topological), geometric problems often require solution multiplicity, motivating the generative extension.

## 2.2. Neural Fields

A *neural field* (NF) (also coordinate-based NN, implicit neural representation (INR)) is a NN (typically a multilayer-perceptron) representing a function $f : x \mapsto y$ that maps a spatial and/or temporal coordinate $x$ to a quantity $y$. Compared to discrete representations, NFs are significantly more memory-efficient while providing higher fidelity, continuity, and access to automatic differentials. They have seen widespread success in representing and generating a variety of signals, including shapes (Park et al., 2019; Chen & Zhang, 2019; Mescheder et al., 2019), scenes (Mildenhall et al., 2021), images (Karras et al., 2021), audio, video (Sitzmann et al., 2020), and physical quantities (Raissi et al., 2019). For a more comprehensive overview, we refer to the survey by Xie et al. (2022).

**Implicit neural shapes** (INSs) represent geometries through scalar fields, such as occupancy (Mescheder et al., 2019; Chen & Zhang, 2019) or signed-distance (Park et al., 2019; Atzmon & Lipman, 2020). In addition to the properties of NFs, INSs also enjoy topological flexibility supporting shape reconstruction and generation. We point out the difference between these two training regimes. In the generative setting, the training is supervised on the ground truth scalar field of every shape. However, in surface reconstruction, i.e., finding a smooth surface from a set of points

measured from a single shape, no ground truth is available and the problem is ill-defined (Atzmon & Lipman, 2020; Berger et al., 2016).

**Regularization** methods have been proposed to counter the ill-posedness in geometry problems. These include leveraging ground-truth normals (Atzmon & Lipman, 2021) and curvatures (Novello et al., 2022), minimal surface property (Atzmon & Lipman, 2021), and off-surface penalization (Sitzmann et al., 2020). A central effort is to achieve the distance field property of the scalar field for which many regularization terms have been proposed: eikonal loss (Gropp et al., 2020), divergence loss (Ben-Shabat et al., 2022), directional divergence loss (Yang et al., 2023), level-set alignment (Ma et al., 2023), or closest point energy (Marschner et al., 2023). The distance field property can be expressed as a PDE constraint called *eikonal equation* $|\nabla f(x)| = 1$, establishing a relation of regularized INSs to PINNs (Gropp et al., 2020).

**Inductive bias.** In addition to explicit loss terms, the architecture, initialization, and optimizer can also limit or bias the learned shapes. For example, typical INSs are limited to watertight surfaces without boundaries or self-intersections (Chibane et al., 2020; Palmer et al., 2022). ReLU networks are limited to piece-wise linear surfaces with sharp vertices and edges and, together with gradient descent, are biased toward low frequencies (Tancik et al., 2020). Fourier-feature encoding (Tancik et al., 2020), sine activations (Sitzmann et al., 2020), and wavelet activations (Saragadam et al., 2023) allow for controlling this frequency bias. Similarly, initialization techniques are important to converge toward desirable optima (Sitzmann et al., 2020; Atzmon & Lipman, 2020; Ben-Shabat et al., 2022; Wang et al., 2023).

## 2.3. (Data-Free) Generative Modeling

Generative modeling (Kingma & Welling, 2013; Goodfellow et al., 2014; Rezende & Mohamed, 2015; Tomczak, 2021) is almost exclusively performed in a data-driven (i.e., supervised) setting to capture and sample from the underlying data distribution. However, notable exceptions exist.

**Boltzmann generators** (Noé et al., 2019) are a prominent example of *data-free* generative models. They are trained to capture the Boltzmann distribution associated with an energy landscape. In the generative setting, GINNs also learn a distribution minimizing an energy as an implicit combination of constraint violations and objectives. However, Boltzmann generators avoid mode-collapse using an entropy-regularizing term, which presupposes invertibility, making them not directly applicable to function spaces. Instead, GINNs use a more general diversity term to hinder mode-collapse over the function space of shapes.

**Conditional neural fields** allow for generative modeling of functions. By conditioning a base network $F$ on a modulation (i.e., latent) variable $z$, a conditional NF encodes multiple fields simultaneously: $f(x) = F(x; z)$. The different choices of the conditioning mechanism lead to a zoo of architectures, including input concatenation (Park et al., 2019), hypernetworks (Ha et al., 2017), modulation (Mehta et al., 2021), and attention (Rebain et al., 2022). These can be classified into global and local mechanisms, which also establishes a connection between conditional NFs and operator learning (Wang et al., 2024). For more detail, we refer to Xie et al. (2022); Rebain et al. (2022).

**Generative design** refers to computational methods that automatically conduct design exploration under constraints set by designers (Jang et al., 2022). It holds the potential of streamlining innovative solutions, e.g., in material design, architecture, or engineering. GINNs can be seen as solving the general task of *topology optimization* – finding the material distribution that minimizes a specified objective subject to constraints. However, while classical methods optimize a single shape directly, we optimize a model that generates diverse feasible shapes. This supports design space exploration and downstream tasks while allowing to incorporate sparse data samples, if available. While generative design datasets are not abundant, deep learning has shown promise in material design and topology optimization. For more detail, we refer to surveys on generative models in engineering design (Regenwetter et al., 2022) and topology optimization via machine learning (Shin et al., 2023).

# 3. Method

Consider the metric space $(\mathcal{F}, d)$ of functions, such as those representing a shape or a PDE solution. Let the set of constraints define the feasible set $\mathcal{K} = \{f \in \mathcal{F} | c_i(f) = 0, i = 1..m\}$. Additionally, let the problem be equipped with an objective $o : \mathcal{F} \mapsto \mathbb{R}$. Selecting the objectives and constraints of a geometric nature lays the foundation for a *GINN*, which is trained to produce an optimal feasible solution by solving $\min_{f \in \mathcal{K}} o(f)$. A key feature of geometric problems is that one is often interested in finding different near-optimal solutions, for example, due to incompleteness, uncertainty, or under-determinedness in the problem specification (e.g., see Figure B.4). This motivates making GINN *generate* a set of sufficiently diverse near-optimal solutions $S \subset \mathcal{K}$:

$$\min_{\substack{S \subset \mathcal{K} \\ \delta(S) \geq \bar{\delta}}} O(S) . \tag{1}$$

$O(S)$ aggregates the objectives $o(f)$ of all solutions $f \in S$ and $\delta$ captures some intuitive notion of *diversity*. While it can be treated as another constraint, it acts on the entire solution set instead of each solution separately. Section 3.1 first discusses representing shapes as functions, in particu-

lar, neural fields, and formulating differentiable constraints. In Section 3.2, we generalize to representing and finding diverse solutions using conditional neural fields.

## 3.1. Finding a Solution

**Representation of a solution.** Let $f : \mathcal{X} \mapsto \mathbb{R}$ be a continuous scalar function on the domain $\mathcal{X} \subset \mathbb{R}^n$. The sign of $f$ *implicitly* defines the shape $\Omega = \{x \in \mathcal{X} | f(x) \leq 0\}$ and its boundary $\partial\Omega = \{x \in \mathcal{X} | f(x) = 0\}$. We use a NN $f = f_\theta$ with parameters $\theta$ to represent the implicit function, i.e., an *implicit neural shape*, due to its memory efficiency, continuity, and differentiability. Nonetheless, the GINNs extend to other representations, as shown in Section 4.2. We additionally require $f$ to approximate the *signed-distance function* (SDF) of $\Omega$ (defined in Eq. 21). This alleviates the ambiguity of many implicit functions representing the same geometry and aids the computation of persistent homology, surface point samples, and diversity. In training, the eikonal constraint is treated analogously to the geometric constraints.

**Constraints on a solution.** To perform gradient-based optimization, we must first ensure each constraint can be written as a differentiable constraint violation $c_i : \mathcal{F} \mapsto \mathbb{R}$. A geometric constraint has the general form $c_i(\Omega, \partial\Omega) = 0$. By representing the shape and the boundary implicitly via the function $f$, the constraints on the sets can be translated into constraints on $f$. This in turn allows to formulate differentiable constraint violations $c_i$, although this choice is not unique. Table 1 shows several examples using the constraints from our experiments. Some losses are straightforward, and some have been previously demonstrated as regularization terms for INSs (see Section 2.2). Section 4.1 discusses two complex losses in more detail: connectedness and smoothness.

## 3.2. Generating Diverse Solutions

**Representation of the solution set.** The *generator* $G : z \mapsto f$ maps a latent variable $z \in Z$ to a function $f$. The solution set is hence the image of the latent set under the generator: $S = \text{Im}_G(Z)$. Furthermore, the generator transforms the input probability distribution $p_Z$ over $Z$ to an output probability distribution $p$ over $S$. In practice, the generator is a modulated base network producing a conditional neural field: $f(x) = F(x; z)$.

**Constraints on the solution set.** By adopting a probabilistic view, we extend each constraint violation and the objective to their expected values: $C_i(S) = \mathbb{E}_{z \sim p_Z} [c_i(G(z)]$ and $O(S) = \mathbb{E}_{z \sim p_Z} [o(G(z)]$.

**Diversity of the solution set.** The last missing piece to training a generative GINN is making $S$ a diverse collection of solutions. In the typical supervised generative modeling

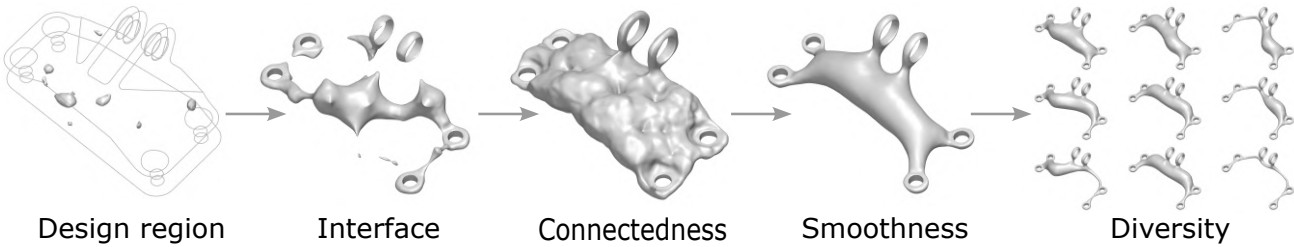

Design region     Interface     Connectedness     Smoothness     Diversity

Figure 3: The user can define geometric problems and solve them using the GINN framework. Here, we illustrate the results of progressively adding more design requirements, overall resulting in a shape generative model trained without data.

| | Set constraint $c_i(\Omega)$ | Function constraint | Constraint violation $c_i(f)$ |
|---|---|---|---|
| Design region | $\Omega \subset \mathcal{E}$ | $f(x) > 0 \,\forall x \notin \mathcal{E}$ | $\int_{\mathcal{X}\setminus\mathcal{E}} [\min(0, f(x))]^2 \, \mathrm{d}x$ |
| Interface | $\partial\Omega \supset \mathcal{I}$ | $f(x) = 0 \,\forall x \in \mathcal{I}$ | $\int_{\mathcal{I}} [f(x)]^2 \, \mathrm{d}x$ |
| Prescribed normal | $n(x) = \bar{n}(x) \,\forall x \in \mathcal{I}$ | $\nabla f(x) = \bar{n}(x) \,\forall x \in \mathcal{I}$ | $\int_{\mathcal{I}} [\nabla f(x) - \bar{n}(x)]^2 \, \mathrm{d}x$ |
| Topology | Using persistent homology; see Section 4.1 and Appendix E | | |

Table 1: Geometric constraints used in several experiments. The shape $\Omega$ and its boundary $\partial\Omega$ are represented implicitly by the (sub-)level set of the function $f$. The shape must be contained within the *design region* $\mathcal{E} \subseteq \mathcal{X}$ and attach to the *interface* $\mathcal{I} \subset \mathcal{E}$ with a prescribed *unit normal* $\bar{n}(x)$. Other interesting constraints are listed in Table 8.

setting, the diversity of the generator is inherited from the diversity of the training dataset. The violation of this is studied under phenomena like *mode-collapse* in GANs (Che et al., 2017). Exploration beyond the training data has been attempted by adding an explicit diversity loss, such as entropy (Noé et al., 2019), Coulomb repulsion (Unterthiner et al., 2018), determinantal point processes (Chen & Ahmed, 2020; Heyrani Nobari et al., 2021), pixel difference, and structural dissimilarity (Jang et al., 2022). We observe that simple generative GINN models are prone to mode-collapse, which we mitigate by adding a *diversity constraint*.

Many scientific disciplines require to measure the diversities of sets which has resulted in a range of definitions of diversity (Parreño et al., 2021; Enflo, 2022; Leinster & Cobbold, 2012). Most start with a *distance* $d : \mathcal{F}^2 \mapsto [0, \infty)$, which can be transformed into the related *dissimilarity*. *Diversity* $\delta : 2^{\mathcal{F}} \mapsto [0, \infty)$ is then the collective dissimilarity of a set (Enflo, 2022), aggregated in some way. In the following, we describe these two aspects: the distance $d$ and the aggregation into the diversity $\delta$.

**Aggregation.** Adopting terminology from Enflo (2022), we use the *minimal aggregation measure*:

$$\delta(S) = \left( \sum_i \left( \min_{j \neq i} d(f_i, f_j) \right)^{1/2} \right)^2 . \quad (2)$$

This choice is motivated by the *concavity* property, which promotes uniform coverage of the available space, as depicted in Figure 13. Figure 5c illustrates how it counteracts mode-collapse in a geometric problem. However,

Equation 2 is well-defined only for finite sets, so, in practice, we apply $\delta$ to a batch of $k$ i.i.d. sampled shapes $S_k = \{G(z_i)|z_1, ..., z_k \overset{iid}{\sim} p_Z\}$. We leave the consideration of diversity on infinite sets, especially with manifold structure, to future research.

**Distance.** A simple choice for measuring the distance between two functions is the $L^2$ function distance $d_2(f_i, f_j) = \sqrt{\int_{\mathcal{X}} (f_i(x) - f_j(x))^2 \, \mathrm{d}x}$. However, recall that we ultimately want to measure the distance between the shapes, not their implicit function representations. For example, consider a disk and remove its central point. While we would not expect their shape distance to be significant, the $L^2$ distance of their SDFs is. This is because local changes in the geometry can cause global changes in the SDF. For this reason, we modify the distance (derivation in Appendix F) to only consider the integral on the shape boundaries $\partial\Omega_i, \partial\Omega_j$ which partially alleviates the globality issue:

$$d(f_i, f_j) = \sqrt{\int_{\partial\Omega_i} f_j(x)^2 \, \mathrm{d}x + \int_{\partial\Omega_j} f_i(x)^2 \, \mathrm{d}x} . \quad (3)$$

If $f_j$ (analogously $f_i$) is an SDF then $\int_{\partial\Omega_i} f_j(x)^2 \, \mathrm{d}x = \int_{\partial\Omega_i} \min_{x' \in \partial\Omega_j} \|x - x'\|_2^2 \, \mathrm{d}x$ and $d$ is closely related to the *chamfer discrepancy* (Nguyen et al., 2021). We note that $d$ is not a metric distance on functions, but recall that we care about the geometries they represent. Using appropriate boundary samples, one may also directly compute a geometric distance, e.g., any point cloud distance (Nguyen et al., 2021).

**In summary**, training a GINN corresponds to solving a constrained optimization problem, i.e. improving the expected objective $O(S)$ and feasibility $C_i(S)$ w.r.t. to each geometric constraint $i = 1..m$ and the diversity constraint $C_{m+1}(S_k) = \max(\delta(S_k) - \bar{\delta}, 0)$. In practice, we convert this into a sequence of unconstrained optimization problems using the augmented Lagrangian method introduced in Section 4.1.

## 4. Experiments

We demonstrate the proposed GINN framework experimentally on a range of problems spanning physics, geometry, and engineering, summarized in a problem-constraint matrix (Table 2). To the best of our knowledge, data-free shape-generative modeling is an unexplored field with no established baselines, problems, and metrics. Thus, in addition to the problems defined and solved in Section 4.2, we define metrics for each constraint as detailed in Appendix C.1. We use these to perform quantitative ablation studies in Appendix C.2 and evaluate baselines in Appendix C.4, reserving the primary text for the main findings. We proceed with an overview of key experimental considerations, with more experimental and implementation details available in Appendix B.

### 4.1. Experimental Details

**Constrained optimization.** To solve the aforementioned constrained optimization problem in Equation 1, we employ the *augmented Lagrangian method* (ALM). It is well studied in the classical and more recently deep learning literature. ALM balances the feasibility and optimality of the solution by controlling the influence of each constraint while avoiding the ill-conditioning and convergence issues of simpler methods. Specifically, we use an *adaptive* ALM (Basir & Senocak, 2023) that uses adaptive penalty parameters $\mu_i$ for each constraint to solve Equation 1 as the unconstrained optimization problem $\max_\lambda \min_\theta \mathcal{L}(\theta, \lambda, \mu)$ where

$$\mathcal{L}(\theta, \lambda, \mu) := O(S_k(\theta)) + \sum_{i=1}^{m+1} \lambda_i C_i(S_k(\theta))$$
$$+ \frac{1}{2} \sum_{i=1}^{m+1} \mu_i C_i^2(S_k(\theta)) . \quad (4)$$

The multipliers $\lambda_i$ and the penalty parameters $\mu_i$ are updated during training according to Equations (18) - (20). Adaptive ALM allows GINNs to handle different constraints without manual hyperparameter tuning for each loss. However, ALM works best if the losses are already on a similar scale. Appendix D provides a more detailed introduction and motivation for this approach.

**Topology** describes properties of a shape that are invariant under deformations, such as the number of holes. Certain materials and objects display specific topological properties (Moore, 2010; Caplan et al., 2018; Bendsoe & Sigmund, 2011), e.g., *connectedness*, which is a basic requirement for the propagation of forces and, by extension, manufacturability and structural function.

Despite topological properties being discrete-valued, *persistent homology* (PH) allows to formulate a differentiable loss. It identifies topological features (e.g., connected components) and quantifies their *persistence* w.r.t. some *filtration* function. For our implicit shapes, this is the implicit function $f$ itself. Consequently, the *birth* and *death* of each feature can be matched to a pair of critical points of $f$. Their values can then be adjusted to achieve the desired topology. In practice, we follow the standard procedure of first discretizing the continuous function onto a cubical complex. We filter cells outside the design region $\mathcal{E}$ to prevent invalid connections. In some experiments, we use an additional constraint that minimizes the number of holes. We detail PH and our approach in Appendix E.

**Smoothness** is another computationally non-trivial design requirement that we consider. Many alternative smoothing energies exist, each leading to different surface qualities (Westgaard & Nowacki, 2001; Song, 2021), but a broad class of smoothing energies can be written as the surface integral $\int_{\partial\Omega \setminus \mathcal{I}} e(\kappa_1(x), \kappa_2(x)) \, dx$ of some curvature expression $e : \mathbb{R}^2 \mapsto \mathbb{R}$. The principal curvatures $\kappa_1$ and $\kappa_2$, same as other differential-geometric quantities, can be computed from $\nabla_x f$ and $H_x f$ in closed-form (Goldman, 2005). To solve Plateau's problem, we use the *mean curvature* $\kappa_H := \frac{\kappa_1 + \kappa_2}{2}$. In the bracket experiment, we use the *surface-strain* $E := \kappa_1^2 + \kappa_2^2$ and a variant thereof $E_{\log} := \log(1 + E)$.

**Surface sampling** is required to estimate the surface integrals for smoothness and diversity. We describe our sampling strategy in Appendix B, noting that it gives a lower variance of the losses and a better convergence compared to a naive strategy.

**Models.** Across the experiments, we consider several NF models that primarily differ in their activation function, including softplus, SIREN (Sitzmann et al., 2020), and WIRE (Saragadam et al., 2023). We require all models to have well-defined and non-vanishing first and second derivatives $\nabla_x f$ and $H_x f$ to compute the iso-level normals and curvatures. As the NF conditioning mechanism, we always use input concatenation (see Section 2.3), denoting the latent space dimension as $\dim(z)$. We continue the model choice discussion in Appendix B.

### 4.2. Problems

**Generative PINN solving an under-determined PDE to demonstrate the generality of the approach.** Although

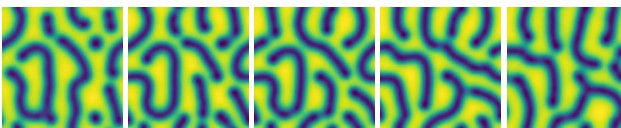

Figure 4: A generative PINN producing Turing patterns that morph during latent space interpolation. This is a result of searching for diverse solutions to an under-determined PDE.

we focus mainly on geometric tasks, we show that the idea of a diversity-constrained data-free neural field generator generalizes to other areas, such as physics. While most familiar problems in physics are well-defined, cases exist where, e.g., the initial conditions are irrelevant and general PDE solutions are sought, such as in chaotic systems or animations. Here, we seek stationary solutions to a *reaction-diffusion* system with *no initial condition*. Note that this system has infinitely many solutions. Figure 4 illustrates the resulting Turing patterns that continuously morph during latent space traversal of the trained generative PINN. Appendix B.1 describes the problem in greater detail.

**Plateau's problem to demonstrate GINNs on a well-posed problem.** Plateau's problem is to find the surface $M$ with the minimal area given a prescribed boundary $\Gamma$ (a closed curve in $\mathcal{X} \subset \mathbb{R}^3$). A *minimal surface* is known to have zero mean curvature $\kappa_H$ everywhere. Minimal surfaces have boundaries and may contain intersections and branch points (Douglas, 1931) which cannot be represented implicitly. For simplicity, we select a suitable problem instance, noting that more appropriate geometric representations exist (Wang & Chern, 2021; Palmer et al., 2022). Altogether, we represent the surface as $M = \partial\Omega \cap \mathcal{X}$ and the two constraints are $\Gamma \subset M$ and $\kappa_H(x) = 0 \ \forall x \in M$. The result in Figure 5a approximates the known solution.

**Parabolic mirror to demonstrate a different geometry representation.** Although we mainly focus on the implicit representation, the GINN framework extends to other representations, such as explicit, parametric, or discrete shapes. Here, the GINN learns the explicit height function $f : [-1, 1] \mapsto \mathbb{R}$ of a mirror with the interface constraint $f(0) = 0$ and that all the reflected rays should intersect at the focal point $(0, 1)$. The result in Figure 5b approximates the known solution: a parabolic mirror. This is a very basic example of caustics, an inverse problem in optics, which we hope inspires future work leveraging the recent advancements in neural rendering techniques.

**Obstacle to introduce connectedness and diversity constraints.** Consider a 2D rectangular design region $\mathcal{E}$ with a circular obstacle in the middle. The interface $\mathcal{I}$ consists of two vertical line segments and has prescribed outward-facing normals $\bar{n}$. We seek shapes that *connect* these two interfaces while avoiding the obstacle. Despite this problem

admitting infinitely many solutions, the naive application of the generative softplus-MLP with $\dim(z) = 1$ leads to mode-collapse. This is mitigated by employing the additional diversity constraint as illustrated in Figure 5c.

**Wheels as a simple design problem.** Consider the domain $X = [-1, 1]^2$ containing the ring-shaped design region $\mathcal{E} = \{x \in X | 0.2^2 \leq x_1^2 + x_2^2 \leq 0.8^2\}$ and the interface $\mathcal{I} = \partial\mathcal{E}$. As before, the shapes must also satisfy the connectedness and diversity constraints. Additionally, a 5-fold cyclic symmetry constraint is required. We implement this as a soft constraint by sampling a point, rotating it by $\frac{2}{5}\pi$ four times, and minimizing the variance of the implicit function $f$ at these five points. Alternatively, exact symmetry can be imposed using a periodic encoding of the input. The result of traversing the 2D latent space of discovered shapes is shown in Figure 5d (larger version in Figure 7).

**Jet engine bracket to demonstrate scaling to 3D engineering design problems.** The problem specification is based on an engineering design competition hosted by General Electric and GrabCAD (Kiis et al., 2013). The challenge was to design the lightest possible lifting bracket of a jet engine subject to both physical and geometrical constraints. Here, we focus only on the geometric requirements: the shape must fit in the given freeform design region $\mathcal{E}$ and attach to five cylindrical interfaces $\mathcal{I}$: a horizontal loading pin and four vertical fixing bolts (see the sketch in Figure 1). Instead of minimizing the volume subject to a linear elasticity PDE constraint, we minimize the surface smoothness $E$ subject to a topological connectedness constraint. Conceptually, this formulation is similar but avoids a PDE solver in the training loop, which we address in a follow-up work. These requirements and several GINN solutions are illustrated in Figures 1 and 3, which also explore additional requirements, including the modified surface energy $E_{\log}$, number of holes, and diversity. The result of latent space traversal of the GINN model trained with diversity is illustrated in Figure 6. In all cases, GINNs produce smooth, singly connected shapes that attach to the interfaces while remaining within the given design space. These properties are quantified in Appendix C.

### 4.3. Discussion

**Solution diversity, generalization, and latent structure.** With the diversity constraint, GINNs not only produce multiple solutions, but we also observe the emergence of a *latent space structure*. This is best seen in Figure 6 using a 2D latent space from which $k = 9$ random samples are drawn every training iteration. Traversing the latent space of the trained GINN produces continuously morphing feasible shapes, i.e., the model *generalizes*. Furthermore, the latent space is *organized* – the solutions vary consistently over large latent distances, and latent directions account

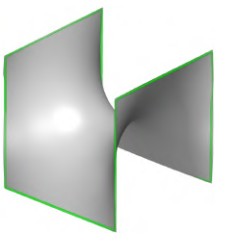

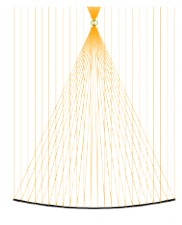

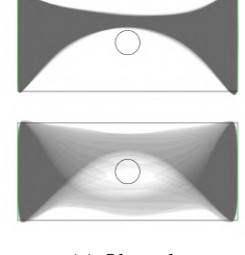



(a) Plateau's problem (b) Parabolic mirror (c) Obstacle (d) Wheels

Figure 5: GINN applied to four different problems. (a) Plateau's problem: the unique minimal surface that attaches to the prescribed boundary. (b) Parabolic mirror: the unique curve that collects reflected rays into a single point. (c) Obstacle: connecting the two interfaces within the allowed design region. A superposition of 16 solutions is shown trained with (bottom) and without (top) a diversity loss, which is required to avoid mode-collapse. (d) Wheel: The diverse and structured latent space of a GINN trained to discover cyclically symmetric wheel designs. A larger version is available in Figure 7.

for different aspects. We see similar behavior in all four experiments that include the diversity constraint. However, we find that the learned structure depends on the exact setup of the diversity constraint. In particular, we observe a more pronounced organization emerge for larger $\bar{\delta}$, however, over-specifying it impedes convergence.

**User control.** The variations of the bracket problem illustrated in Figure 1 highlight the user's ability to tune the problem and the resulting solutions. We solve these with the same setup and hyperparameters, illustrating the robustness of GINNs and the adaptive ALM approach to constrained optimization.

**Optimization.** A representative convergence behavior of the training is shown in Figures 11 and 12. Despite up to seven loss terms, adaptive ALM automatically balances these and minimizes each constraint violation. However, the variance in several losses remains high. This is largely due to the diversity and smoothness terms, which are hard to optimize and increase the necessary number of iterations by roughly a factor of two and five, respectively. Curvature is a second-order differential operator and is expected to be ill-conditioned, motivating the future use of second-order optimizers (Ryck et al., 2024; Rathore et al., 2024).

**Runtime** of a single bracket shape is roughly 10K iterations or 30 minutes. Similarly, the diverse model trains for around 50K iterations or 5 hours on a single GPU. Of the total time, the surface strain takes roughly 10% (due to the Hessian) and the PH solver 75% (expensive multi-processed CPU task). The runtime also increases when ablating the eikonal constraint as it destroys the geometric regularity of the implicit function, hindering efficient surface point sampling that usually takes 15% of total time. More details are available in Appendix C.3.

**Ablations** are performed in Appendix C.2, measuring the expected role of each constraint, as well as ALM. Less

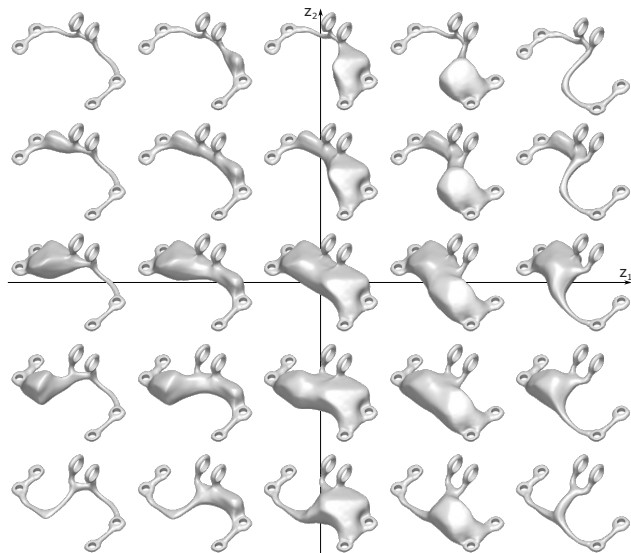

Figure 6: With the diversity constraint, GINNs not only produce multiple solutions but also discover a *latent space structure*. Traversing the 2D latent space morphs solutions, i.e., the model *generalizes*. The latent space is also *organized* – a central bulky shape becomes thinner in the radial direction, and the axes can be identified by how the shape connects on the sides. Figure 9 shows a large $9 \times 9$ version.

obvious is the strong impact of the eikonal constraint on the connectedness, smoothness, and diversity metrics, again due to the lack of geometric regularity of the implicit function. A progressive ablation of all requirements of the bracket problem is illustrated in Figure 3. Figure 8 also visually compares different NF models, highlighting the superiority of WIRE over softplus-MLP and SIREN due to their spectral behavior.

# 5. Conclusion

We have introduced geometry-informed neural networks demonstrating shape-generative modeling driven solely by geometric constraints and objectives. After formulating the learning problem and discussing key theoretical and practical aspects, we applied GINNs to a range of problems.

**Limitations and future work.** GINNs combine several known and novel components, each of which warrants an in-depth study of theoretical and practical aspects, including alternative shape distances, their aggregation into diversity, conditioning mechanisms, constraints, and optimization.

In this work, we focused on building the conceptual framework of GINNs and validating it experimentally. This included a realistic engineering design task. However, we considered a modified version of the original task and did not compare to established topology-optimization methods as this required the integration of a PDE solver – a task we address in a follow-up work.

Even though ALM is a significant improvement over the naive approach of manually weighted loss terms, the recent literature on multi-objective and second-order optimizers suggests further possible improvements.

Finally, we investigated GINNs in the limit of no data. However, GINNs can integrate partial observations of a single or multiple shapes. This combination of classical and machine learning methods suggests a new approach to generative design in data-sparse settings, which are of high relevance in practical engineering settings.

# Acknowledgments

We sincerely thank Georg Muntingh and Oliver Barrowclough for their feedback on the paper.

The ELLIS Unit Linz, the LIT AI Lab, the Institute for Machine Learning, are supported by the Federal State Upper Austria. We thank the projects FWF AIRI FG 9-N (10.55776/FG9), AI4GreenHeatingGrids (FFG- 899943), Stars4Waters (HORIZON-CL6-2021-CLIMATE-01-01), FWF Bilateral Artificial Intelligence (10.55776/COE12). We thank the European High Performance Computing initiative for providing computational resources (EHPC-DEV-2023D08-019, 2024D06-055, 2024D08-061). We thank NXAI GmbH, Audi AG, Silicon Austria Labs (SAL), Merck Healthcare KGaA, GLS (Univ. Waterloo), TÜV Holding GmbH, Software Competence Center Hagenberg GmbH, dSPACE GmbH, TRUMPF SE + Co. KG.

Arturs Berzins was supported by the European Union's Horizon 2020 Research and Innovation Programme under Grant Agreement number 860843.

# Impact Statement

Our work aims to advance the field of machine learning and may contribute to its broader societal impact. In addition, there is an ongoing discussion on the rights to data, since data is fundamental to training the majority of current machine learning models. The demonstrated data-free approach to generative modeling brings forth a less explored perspective on this matter. It circumvents the copyright problem and facilitates practitioners who lack exclusive access to datasets. However, for the foreseeable future, the applicability and hence the impact is limited to scientific and engineering applications. The demonstrated results, in particular, might foster a path toward improved approaches to engineering design.

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

## A. Problem-Constraint Matrix

Table 2 provides an overview of the variety of considered tasks using a problem-constraint matrix. Not captured by this matrix are also the differences in domain dimension, shape representation, and problem symmetries.

Table 2: Problem-constraint matrix.

|  | Turing patterns | Plateau's problem | Parabolic mirror | Obstacle | Wheels | Jet engine bracket |
|---|---|---|---|---|---|---|
| Reaction-diffusion PDE | + |  |  |  |  |  |
| Interface |  | + | + | + | + | + |
| Mean-curvature |  | + |  |  |  |  |
| Reflection |  |  | + |  |  |  |
| Design region |  |  |  | + | + | + |
| Prescribed normal |  |  |  | + | + | + |
| Connectedness |  |  |  | + | + | + |
| Diversity | + |  |  | + | + | + |
| Eikonal |  |  |  | + | + | + |
| Rotational symmetry |  |  |  |  | + |  |
| Holes |  |  |  |  |  | + |
| Surface strain |  |  |  |  |  | + |
| Minimum thickness |  |  |  |  | + | + |

## B. Implementation and Experimental Details

We report additional details on the experiments and their implementation. We run all experiments on a single GPU (one of NVIDIA RTX2080Ti, RTX3090, A40, P40, or A100-SXM). The largest GPU memory requirement is 16GB for the multi-shape training of the jet engine bracket (9 shapes).

**Surface sampling** is required to estimate the surface integrals for smoothness and diversity. We first sample points in the envelope and project them onto the surface using Newton iterations. We then repel the points on the surface to achieve a more uniform distribution similar to Yifan et al. (2020). Finally, we exclude points sampled within a small distance to the interface $\mathcal{I}$, as the surface should not change here. We also begin to sample surface points and compute the surface integrals only after a warm-up phase of 500 iterations. In combination, these aspects lead to a lower variance and better convergence.

### B.1. Reaction-Diffusion

The high-level idea of GINNs is to find diverse solutions to an under-determined problem. While the main focus of the paper is on geometry, we show that this idea generalizes to other areas. In physics, problems are often well-defined and have a unique solution. However, cases exist where the initial conditions are irrelevant and a non-particular PDE solution is sufficient, such as in chaotic systems or animations.

We demonstrate an analogous concept of *generative PINNs* on a *reaction-diffusion* system. Such systems were introduced by Turing (1952) to explain how patterns in nature, such as stripes and spots, can form as a result of a simple physical process of reaction and diffusion of two substances. A celebrated model of such a system is the Gray-Scott model (Pearson, 1993), which produces a variety of patterns by changing just two parameters – the feed-rate $\alpha$ and the kill-rate $\beta$ – in the following PDE:

$$\frac{\partial u}{\partial t} = D_u \Delta u - uv^2 + \alpha(1 - u) , \quad \frac{\partial v}{\partial t} = D_v \Delta v + uv^2 - (\alpha + \beta)v . \tag{5}$$

This PDE describes the concentration $u, v$ of two substances $U, V$ undergoing the chemical reaction $U + 2V \rightarrow 3V$. The rate of this reaction is described by $uv^2$, while the rate of adding $U$ and removing $V$ is controlled by the parameters $\alpha$ and $\beta$. Crucially, both substances undergo diffusion (controlled by the coefficients $D_u, D_v$), which produces an instability leading to rich patterns around the bifurcation line $\alpha = 4(\alpha + \beta)^2$.

Computationally, these patterns are typically obtained by evolving a given initial condition $u(x, t = 0) = u_0(x)$, $v(x, t = 0) = v_0(x)$ on some domain with periodic boundary conditions. A variety of numerical solvers can be applied, but previous PINN attempts fail without data (Giampaolo et al., 2022). To demonstrate a generative PINN on a problem that admits multiple solutions, we omit the initial condition and instead consider stationary solutions, which are known to exist for some parameters $\alpha, \beta$ (McGough & Riley, 2004). We use the corresponding stationary PDE ($\partial u/\partial t = \partial v/\partial t = 0$) to formulate the residual losses:

$$L_u = \int_{\mathcal{D}} (D_u \Delta u - uv^2 + \alpha(1 - u))^2 \, \mathrm{d}x \,, \quad L_v = \int_{\mathcal{D}} (D_v \Delta v + uv^2 - (\alpha + \beta)v)^2 \, \mathrm{d}x \,. \tag{6}$$

To avoid trivial (i.e. uniform) solutions, we encourage non-zero gradient with a loss term $- \max(1, \int_{\mathcal{D}} (\nabla u(x))^2 + (\nabla v(x))^2 \, \mathrm{d}x)$. We find that architecture and initialization are critical (described below). Using the diffusion coefficients $D_v = 1.2 \times 10^{-5}$, $D_u = 2D_v$ and the feed and kill-rates $\alpha = 0.028, \beta = 0.057$, the generative PINN produces diverse and smoothly changing pattern of worms, illustrated in Figure 4. To the best of our knowledge, this is the first PINN that produces 2D Turing patterns in a data-free setting.

**Experimental details.** We use two identical SIREN networks for each of the fields $u$ and $v$. They have two hidden layers of widths 256 and 128. We enforce periodic boundary conditions on the unit domain $\mathcal{X} = [0, 1]^2$ through the encoding $x_i \mapsto (\sin 2\pi x_i, \cos 2\pi x_i)$ for $i = 1, 2$. With this encoding, we use $\omega_0 = 3.0$ to initialize SIREN. We also find that the same shaped Fourier-feature network (Tancik et al., 2020) with an appropriate initialization of $\sigma = 3$ works equally well. We compute the gradients and the Laplacian using finite differences on a $64 \times 64$ grid, which is randomly translated in each epoch. Automatic differentiation produces the same results for an appropriate initialization scheme, but finite differences are an order of magnitude faster. The trained fields $u, v$ can be sampled at an arbitrarily high resolution without displaying any artifacts. The generative PINNs are trained with Adam for 20000 epochs with a $10^{-3}$ learning rate taking a few minutes.

## B.2. Plateau's Problem

The model is an MLP with $[3, 256, 256, 256, 1]$ neurons per layer and the $\tanh$ activation. We train with Adam (default parameters) for 10,000 epochs with a learning rate of $10^{-3}$, taking around three minutes. The three losses (interface, mean curvature, and eikonal) are weighted equally, but the mean curvature loss is introduced only after 1000 epochs. To facilitate a higher level of detail, the corner points of the prescribed interface are weighted higher.

## B.3. Parabolic Mirror

The model is an MLP with $[2, 40, 40, 1]$ neurons per layer and the $\tanh$ activation. We train with Adam (default parameters) for 3000 epochs with a learning rate of $10^{-3}$, taking around ten seconds.

## B.4. Obstacle

The obstacle experiment serves as a proof of concept for including several losses, in particular the connectedness loss.

**Problem definition.** Consider the domain $\mathcal{X} = [-1, 1] \times [-0.5, 0.5]$ and the design region that is a smaller rectangular domain with a circular obstacle in the middle: $\mathcal{E} = ([-0.9, 0.9] \times [-0.4, 0.4]) \setminus \{x_1^2 + x_2^2 \le 0.1^2\}$. There is an interface consisting of two vertical line segments $\mathcal{I} = \{(\pm 0.9, x_2)| - 0.4 \le x_2 \le 0.4\}$ with the prescribed outward facing normals $\bar{n}(\pm 0.9, -0.4 \le x_2 \le 0.4) = (\pm 1, 0)$.

**Softplus-MLP.** The neural network model $f$ should be at least twice differentiable with respect to the inputs $x$, as necessitated by the computation of surface normals and curvatures. Since the second derivatives of an ReLU MLP are zero everywhere, we use the softplus activation function as a simple baseline. In addition, we add residual connections (Dugas et al., 2000) to mitigate the vanishing gradient problem and facilitate learning. We denote this architecture with "softplus-MLP". We train a softplus-MLP with $4 \times 512$ hidden layers with Adam (default settings).

**Conditioning the model.** For training the conditional models, we approximate the one-dimensional latent set $Z = [-1, 1]$ with $N = 16$ fixed equally spaced samples. This enables the reuse of some calculations across epochs and results in a well-structured latent space, illustrated through latent space interpolation in Figure 5c.

**Computational cost.** The total training wall-clock time is around 10 minutes for a single shape and approximately 60 minutes for 16 shapes.

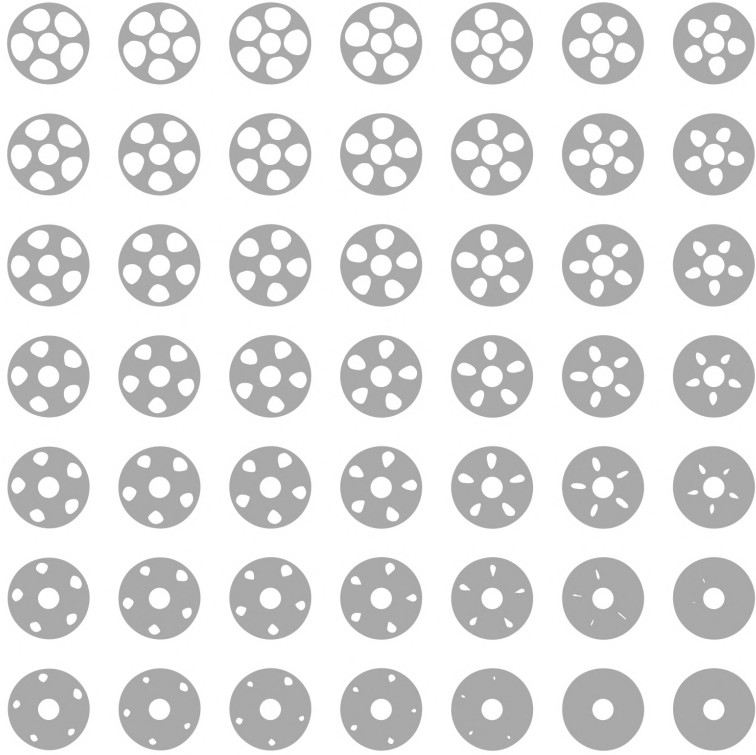

Figure 7: GINNs produce diverse shapes with a structured latent space. The shapes morph continuously into one another when traversing the 2D latent space. From the top-left to bottom-right, the holes become smaller, while from bottom-left to top-right, the holes move inwards.

## B.5. Wheel

The problem setup is as described in the main text. The ML setup is shared with the bracket problem. A larger illustration of the results is available in Figure 7, showing diverse shapes and a structured latent space.

## B.6. Jet Engine Bracket

The jet engine bracket (JEB) is our most complex experiment. We tested different architectures (c.f. Figure 8) and found that WIRE (Saragadam et al., 2023) produced the best results, while being easier to train with ALM than softplus-MLP or SIREN (Sitzmann et al., 2020). We train the WIRE with $3 \times 128$ hidden layers with Adam (default settings) and a learning rate scheduler $0.5^{t/10000}$ for $t = 10000$ iterations for the single shape and $t = 50000$ for multiple shapes. To decrease the training time, we use multi-processing to asynchronously create diagnostic plots or computing the PH loss for multiple shapes.

**WIRE.** For the jet engine bracket settings, early experiments indicated that the softplus-MLP cannot satisfy the given constraints. We therefore employ a WIRE network (Saragadam et al., 2023), which is biased towards higher frequencies of the input signal. As mentioned by the authors, the spectral properties of a WIRE model are relatively robust. Several values for $\omega_0$ and $s_0$, which control the frequency and scale of the Gaussian of the first layer at initialization, were tested. As there was no big difference in the results, we fixed them to $\omega_0 = 18$ and $s_0 = 6$ For more detailed results, we refer to Section C.

**Conditioning the model.** In the generative GINN setting, we condition WIRE using input concatenation which can be interpreted as using different biases at the first layer. As we refer in the main text, we leave more sophisticated conditioning techniques for future work. We use $N = 9$ different latent codes spaced in the interval $Z = [0, 0.1]$ and are resampled every iteration. The results are shown in Figure 9.

**Spatial resolution.** The curse of dimensionality implies that with higher dimensions, exponentially (in the number of

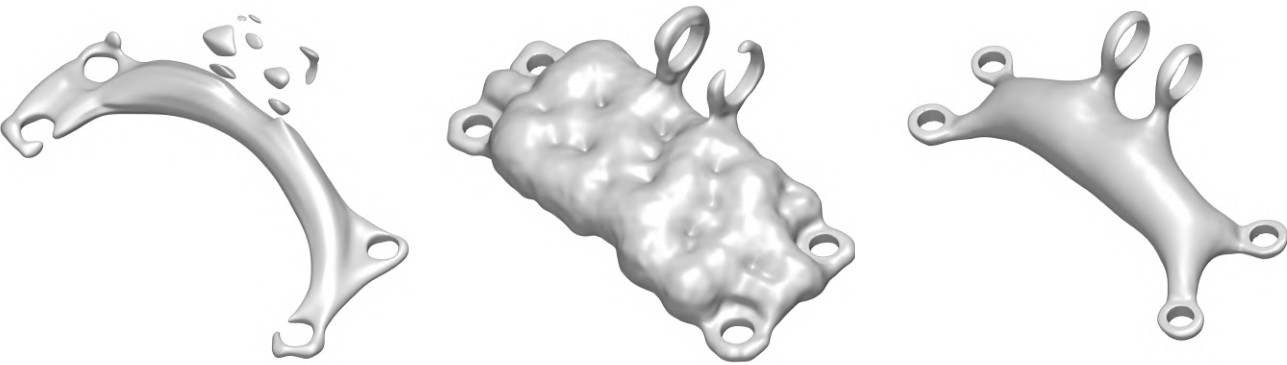

Figure 8: Comparison of architectures trained for 10k epochs to produce a single shape. From left to right: softplus-MLP, SIREN, WIRE. The softplus-MLP is unable to fit the interfaces due to the low-frequency bias. SIREN converges much slower than WIRE, especially at the interfaces, and does not produce a smooth shape.

Figure 9: GINNs produce diverse shapes with a structured latent space. The shapes morph continuously into one another when traversing the 2-dimensional latent space. These shapes are produced by the same model as Figure 6. A trained GINN allows the user to sample densely in the latent space with shapes all meeting the constraints: Interfaces are modeled correctly, shapes are not disconnected or leave the design space.

dimensions) more points are needed to cover the space equidistantly. Therefore, in 3D, substantially more points (and consequently memory and compute) are needed than in 2D. In our experiments, we observe that a low spatial resolution around the interfaces prevents the model from learning high-frequency details, likely due to a stochastic gradient. Increased spatial resolution results in a better learning signal, and the model picks up the details more easily. To facilitate learning, we additionally increase the resolution around the interfaces.

## C. Additional Evaluation

After defining the metrics used in our experiments in Appendix C.1, we perform an additional study in Appendix C.2 ablating each constraint, as well as the ALM, for three tasks. Additionally, in Appendix C.3, we measure the impact of each constraint on the runtime. Finally, Appendix C.4 discusses and applies two baseline methods to the jet engine bracket problem.

### C.1. Metrics

We introduce several metrics for each constraint. We will use the chamfer divergence (Nguyen et al., 2021) to compute the divergence measure between two shapes $P$ and $Q$. For better interpretability, we take the square root of the common definition of chamfer divergence

$$CD_1(P,Q) = \sqrt{\frac{1}{|Q|} \sum_{x \in Q} \min_{y \in P} ||x-y||_2^2} \tag{7}$$

and, similarly, for the two-sided chamfer divergence.

$$CD_2(P,Q) = \sqrt{\frac{1}{|Q|} \sum_{x \in Q} \min_{y \in P} ||x-y||_2^2 + \frac{1}{|P|} \sum_{x \in P} \min_{y \in Q} ||x-y||_2^2}\,. \tag{8}$$

Reusing the notation from the main text, let $\mathcal{E}$ be the design region, $\delta\mathcal{E}$ the boundary of the design region, $\mathcal{I}$ the interface consisting of $n_{\mathcal{I}}$ connected components, $\mathcal{X}$ the domain, $\Omega$ the shape and $\delta\Omega$ its boundary. Let $\text{vol}(P) = \int_P dP$ be the generalized volume (i.e., length, area, or volume) of $P$.

**Design region.** We introduce two metrics to quantify how well a shape fits the design region. Intuitively, for 3D, the first metric quantifies how much volume is outside the design region $\mathcal{E}$ compared to the overall available volume. The second metric compares how much surface area intersects the boundary of the design region. For both, the optimal values are $0$, with lower being better.

- $\frac{\text{vol}(\Omega \setminus \mathcal{E})}{\text{vol}(\mathcal{X} \setminus \mathcal{E})}$: The $d$-volume (i.e. volume for $d=3$ or area for $d=2$) outside the design region, divided by the total $d$-volume outside the design region.

- $\frac{\text{vol}(\Omega \cap \delta\mathcal{E})}{\text{vol}(\delta\mathcal{E})}$: The $(d-1)$-volume (i.e. the surface area for $d=3$ or length of contours for $d=2$) of the shape intersected with the design region boundary, normalized by the total $(d-1)$-volume of the design region.

**Interface.** To measure the goodness of fit to the interface, we use the *one-sided* chamfer distance of the boundary of the shape to the interface, as we do not care if some parts of the shape boundary are far away from the interface, as long as there are some parts of the shape which *are* close to the interface. The best fit is $0$, with lower being better.

- $CD_1(\Omega, \mathcal{I})$: The average minimal distance from sampled points of the interface to the shape boundary.

**Connectedness.** For the connectedness, we care whether the shape itself and the interfaces are connected. Since the shape could potentially connect through paths outside the design region, we also introduce a connectedness metric that accounts for this undesirable effect. $DC(\Omega)$ denotes all connected components of a shape $\Omega$ except the largest. We define the metrics as follows:

- $b_0(\Omega)$: The zeroth Betti number represents the number of connected components of the shape. The target in our work is always 1.

- $b_0(\Omega \cap \mathcal{E})$: The zeroth Betti number of the shape restricted to the design region.

- $\frac{\text{vol}(DC(\Omega))}{\text{vol}(\mathcal{E})}$: The normalized $d$-volume (i.e. volume for $d = 3$ and area for $d = 2$) of disconnected components.

- $\frac{\text{vol}(DC(\Omega \cap \mathcal{E}))}{\text{vol}(\mathcal{E})}$: The normalized $d$-volume of disconnected components *inside the design region*.

- $\frac{CI(\Omega, \mathcal{I})}{n_{\mathcal{I}}}$: The share of connected interfaces. If an interface is an $\epsilon$-distance from a connected component of a shape, we consider it connected to the shape. This metric then represents the maximum number of connected interfaces of any connected component, divided by the total number of interface components. By default, we set $\epsilon = 0.01$ when the domain bounds are comparable to the unit cube.

**Diversity.** We define the diversity $\delta_{\text{mean}}$ on a finite set of shapes $S = \{\Omega_i, i \in [N]\}$ as follows:

$$
\delta_{\text{mean}}(S) = \left[ \frac{1}{N} \sum_{i \in [N]} \left( \frac{1}{N-1} \sum_{j \neq i \in [N]} CD_2(\Omega_i, \Omega_j) \right)^{\frac{1}{2}} \right]^2.
\tag{9}
$$

**Smoothness.** There are many choices of smoothness measures in multiple dimensions. In this paper, we use a Monte Carlo estimate of the *surface strain* (Goldman, 2005) (also mentioned in Section 4). To make the metric more robust to large outliers (e.g., tiny disconnected components have very large curvature and surface strain), we clip the surface strain of a sampled point $x_i, i \in [N]$ with a value $\kappa_{\text{max}} = 1,000,000$.

$$
E_{\text{strain}}(\Omega) = \frac{1}{N} \sum_{i \in [N]} \min \left[ \text{div}^2 \left( \nabla \frac{f(x_i)}{|f(x)|} \right), \kappa_{\text{max}} \right]
\tag{10}
$$

## C.2. Ablations

Using the metrics defined above, the impact of ablating each constraint, as well as ALM, is reported for the obstacle (Table 3), wheel (Table 4), and bracket (Table 5) tasks.

Table 3: Metrics for the ablations of the obstacle problem for multiple shape solutions.

| Ablation | Connectedness | | | | Interface | | Design region | Diversity |
|---|---|---|---|---|---|---|---|---|
| | $\downarrow b_0(\Omega)$ | $\downarrow b_0(\Omega \cap E)$ | $\downarrow \frac{\text{vol}(DC(\Omega))}{\text{vol}(E)}$ | $\downarrow \frac{\text{vol}(\Omega \setminus E)}{\text{vol}(X \setminus E)}$ | $\uparrow \frac{CI(\Omega, I)}{n_{\mathcal{I}}}$ | $\downarrow CD_1(\Omega, I)$ | $\downarrow \frac{\text{vol}(\Omega \cap \delta E)}{\text{vol}(\delta E)}$ | $\uparrow \delta_{\text{mean}}$ |
| (None) | 1.11 | 1.11 | 0.02 | 0.00 | 0.39 | 0.03 | 0.01 | 0.15 |
| Interface | 0.00 | 1.00 | 0.00 | 0.00 | 0.00 | 0.00 | 0.00 | 0.00 |
| Design region | 1.00 | 1.00 | 0.00 | 0.53 | 2.00 | 0.00 | 0.71 | 0.08 |
| Prescribed normal | 0.00 | 1.00 | 0.00 | 0.00 | 0.00 | 0.00 | 0.00 | 0.00 |
| Connectedness | 0.56 | 1.00 | 0.00 | 0.00 | 0.00 | 0.00 | 0.00 | 1.45 |
| Diversity | 1.22 | 1.22 | 0.02 | 0.00 | 0.15 | 0.03 | 0.00 | 0.12 |
| Surface strain | 1.11 | 1.11 | 0.01 | 0.00 | 0.22 | 0.03 | 0.00 | 0.15 |
| Augmented Lagrangian | 1.22 | 1.22 | 0.02 | 0.00 | 0.28 | 0.03 | 0.01 | 0.14 |

Table 4: Metrics for the ablations of the wheel problem for multiple shape solutions.

| Ablation | Connectedness | | | | Interface | | Design region | Diversity |
|---|---|---|---|---|---|---|---|---|
| | $\downarrow b_0(\Omega)$ | $\downarrow b_0(\Omega \cap E)$ | $\downarrow \frac{\text{vol}(DC(\Omega))}{\text{vol}(E)}$ | $\downarrow \frac{\text{vol}(\Omega \setminus E)}{\text{vol}(X \setminus E)}$ | $\uparrow \frac{CI(\Omega,I)}{n_\mathcal{I}}$ | $\downarrow CD_1(\Omega,I)$ | $\downarrow \frac{\text{vol}(\Omega \cap \delta E)}{\text{vol}(\delta E)}$ | $\uparrow \delta_{\text{mean}}$ |
| (None) | 1.11 | 1.00 | 0.00 | 0.00 | 1.00 | 0.00 | 0.07 | 0.31 |
| Interface | 1.00 | 1.00 | 0.00 | 0.00 | 0.44 | 0.07 | 0.02 | 1.10 |
| Design region | 6.89 | 1.00 | 0.06 | 0.11 | 1.00 | 0.00 | 0.44 | 0.68 |
| Prescribed normal | 1.00 | 1.00 | 0.00 | 0.00 | 1.00 | 0.00 | 0.03 | 0.19 |
| Connectedness | 1.22 | 1.22 | 0.02 | 0.00 | 0.89 | 0.00 | 0.11 | 0.34 |
| Diversity | 1.00 | 1.00 | 0.00 | 0.00 | 1.00 | 0.00 | 0.08 | 0.03 |
| Rotational symmetry | 1.00 | 1.00 | 0.00 | 0.00 | 1.00 | 0.00 | 0.06 | 0.54 |
| Minimum thickness | 1.11 | 1.00 | 0.00 | 0.00 | 1.00 | 0.00 | 0.10 | 0.38 |
| Eikonal | 1.00 | 1.00 | 0.00 | 0.00 | 1.00 | 0.00 | 0.09 | 0.03 |
| Curvature & Diversity | 1.00 | 1.00 | 0.00 | 0.00 | 1.00 | 0.00 | 0.08 | 0.03 |
| Augmented Lagrangian | 1.00 | 1.00 | 0.00 | 0.00 | 1.00 | 0.00 | 0.01 | 0.21 |

Table 5: Metrics for ablations of the jet engine bracket problem for single and multiple shape solutions.

| Ablation | Connectedness | | | | Interface | | Design region | Smoothness | Diversity |
|---|---|---|---|---|---|---|---|---|---|
| | $\downarrow b_0(\Omega)$ | $\downarrow b_0(\Omega \cap E)$ | $\downarrow \frac{\text{vol}(DC(\Omega))}{\text{vol}(E)}$ | $\downarrow \frac{\text{vol}(\Omega \setminus E)}{\text{vol}(X \setminus E)}$ | $\uparrow \frac{CI(\Omega,I)}{n_\mathcal{I}}$ | $\downarrow CD_1(\Omega,I)$ | $\downarrow \frac{\text{vol}(\Omega \cap \delta E)}{\text{vol}(\delta E)}$ | $\downarrow E_{\text{strain}}(\Omega)$ | $\uparrow \delta_{\text{mean}}$ |
| **Multiple shapes** | | | | | | | | | |
| (None) | 1.00 | 1.00 | 0.00 | 0.00 | 1.00 | 0.00 | 0.00 | 410 | 0.14 |
| Interface | 1.11 | 1.11 | 0.00 | 0.00 | 1.00 | 0.02 | 0.00 | 373 | 0.13 |
| Design region | 4.44 | 1.00 | -0.01 | 0.97 | 1.00 | 0.00 | 0.14 | 95 | 0.09 |
| Prescribed normal | 1.00 | 1.00 | 0.00 | 0.00 | 1.00 | 0.01 | 0.00 | 343 | 0.13 |
| Connectedness | 25.11 | 19.44 | 0.00 | 0.00 | 0.17 | 0.01 | 0.00 | 178740 | 0.02 |
| Diversity | 1.00 | 1.00 | 0.00 | 0.00 | 1.00 | 0.00 | 0.00 | 341 | 0.05 |
| Surface strain | 1.00 | 1.00 | 0.00 | 0.00 | 1.00 | 0.00 | 0.00 | 263 | 0.10 |
| Minimum thickness | 1.00 | 1.00 | 0.00 | 0.00 | 1.00 | 0.01 | 0.00 | 432 | 0.15 |
| Eikonal | 6.67 | 6.44 | 0.00 | 0.00 | 1.00 | 0.00 | 0.00 | 722 | 0.07 |
| Curvature & Diversity | 1.00 | 1.00 | 0.00 | 0.00 | 1.00 | 0.00 | 0.00 | 305 | 0.05 |
| Augmented Lagrangian | 3.22 | 2.00 | 0.00 | 0.00 | 1.00 | 0.00 | 0.01 | 198 | 0.07 |
| **Single shape** | | | | | | | | | |
| (None) | 1.00 | 1.00 | 0.00 | 0.00 | 1.00 | 0.01 | 0.00 | 291 | |
| Augmented Lagrangian | 1.00 | 1.00 | 0.00 | 0.01 | 1.00 | 0.01 | 0.06 | 170 | |

## C.3. Runtimes

In addition to quantifying the impact of the constraint ablations on the metrics, we also report the iteration times in Table 6, focusing on the 3D jet engine bracket problem, which has the highest number and complexity of constraints. The other experiments show similar behavior. Most constraints have little effect on the time per iteration. Of the total runtime, the surface strain takes 10% (due to the Hessian) and the PH solver 75% (expensive multi-processed CPU task). The runtime increases when ablating the eikonal constraint as it destroys the geometric regularity of the implicit function, hindering efficient surface point sampling that usually takes 15% of total time. We use an A100-SXM GPU, and the peak memory usage for a batch of 9 shapes does not exceed 16 GB. These two losses also have a strong impact on the iterations needed. As discussed in the main text, adding the smoothness loss increases the number of iterations roughly two-fold, and adding the diversity increases it roughly five-fold. Overall, the biggest effect is not so much the number of constraints, but rather losses that are ill-conditioned. This conditioning issue is also known in the PINN literature, and its mitigation is an open and active research topic.

Table 6: Time per iteration when ablating each constraint. Largest differences from the baseline (None) indicate computationally expensive components of the method, most notably the connectedness constraint (due to the CPU-based PH calculation) and the boundary sampler required for calculating both the diversity and surface-strain.

| Ablated constraint | Time [ms] per iteration |
|---|---|
| (None) | 260 |
| Eikonal | 291 |
| Interface | 264 |
| Design region | 262 |
| Prescribed normal | 265 |
| Diversity | 265 |
| Surface strain | 230 |
| Surface strain & Diversity (no boundary point sampler) | 187 |
| Connectedness | 94 |

## C.4. Baselines

While a complete and fair comparison to a baseline is not available, we compare to two at least partially fitting baselines.

**Topology optimization.** We consider classical TO, specifically, FeniTop (Jia et al., 2024) which implements the standard SIMP method with a popular FEM solver. We define a TO problem that is as similar as possible, applying a diagonal force to the top cylindrical pin interface and allowing a 7% volume fraction in the same design region. The other interfaces are fixed in the same way. The shape compliance is minimized for 400 iterations on a $104 \times 172 \times 60$ FEM grid (taking 190 min on a 32 core CPU to give a sense of runtime, although a fair timing comparison requires a more nuanced discussion). The produced shape is visualized in Figure 10.
We then compute the surface strain (the objective we use) for this TO shape and, conversely, the compliance for a GINN shape (Section 4.2; illustrated in the penultimate column of Figure 3). Unsurprisingly, both shapes perform best at the objective they are optimized for while satisfying the constraints up to the relevant precision. This serves as a sanity check and confirmation of the constraint satisfaction.

Table 7: GINN compared to the topology optimization baseline for a single jet engine bracket shape. Both methods perform best on the objectives they are optimized for while satisfying the shared constraints up to the relevant precision.

| Metric | Topology optimization | GINN |
|---|---|---|
| ↓ Connectedness (0-th Betti-number) | 1 | 1 |
| ↓ Interface (Chamfer distance) | 0.00 | 0.00 |
| ↓ Design region (Volume outside) | 0.00 | 0.00 |
| ↓ Curvature | 442 | **144** |
| ↓ Compliance | **0.99** | 0.344 |

**Human-expert dataset.** A unique aspect of GINN is the data-free shape generative aspect. Comparison to classical TO is trivial since it is inherently limited to a single solution with null diversity. Instead, we use the simJEB (Whalen et al., 2021) dataset to give an intuitive estimate of the diversity of the produced results. The dataset is due to the design challenge on a related problem described in Section 4.2. The shapes in the dataset were produced by human experts, many of whom also used topology optimization. To compute the diversity metric, we sample 196 clean shapes from the simJEB dataset, producing a diversity of 0.099, and $14 \times 14$ equidistant samples from the 2D latent space generative GINN model, producing a diversity of 0.167. Even though these sets are not directly comparable as they optimize for different objectives, these results indicate that GINNs can produce diversity on the same and larger magnitude as a dataset that required an estimated collective effort of 14 expert human years (Whalen et al., 2021).

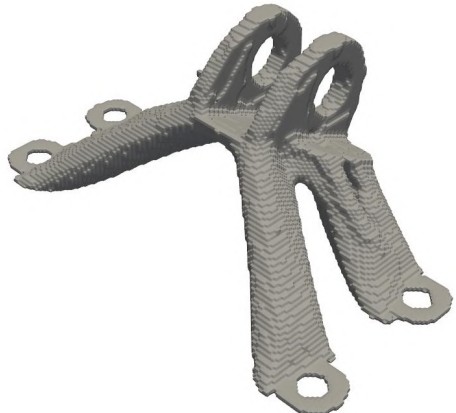

Figure 10: A solution to the jet engine bracket problem found by the FeniTop (Jia et al., 2024) topology optimization software. The quantitative metrics are reported in Table 7.

## D. Optimization

In general, an equality-constrained optimization problem can be written as

$$\min_{\theta} O(\theta) \qquad \text{such that} \qquad C_i(\theta) = 0 \quad \forall i \in 0, \dots, m \qquad (11)$$

where $O, C_1 \dots C_m$ are smooth scalar functions $\mathbb{R}^N \to \mathbb{R}$. $O$ is the *objective function* and *constraint functions* $C_i$ represent the collection of equality constraints. A naive approach to solve this optimization problem is to simply relax the constraints into the objective function and solve the unconstrained optimization problem

$$\min_{\theta} O(\theta) + \mu_{0_k} \sum_{i=0}^{m} C_i(\theta) \qquad (12)$$

for a sequence $\{\mu_{0_k}\}$ with $\mu_{0_k} \leq \mu_{0_{k+1}}$ for all $k$ and $\mu_{0_k} \to \infty$. However, this *penalty method* can suffer from numerical instabilities for large $\mu_{0_k}$, hence the sequence is generally capped at a maximum value $\mu_{max}$. A further problem, which has recently been studied regarding PINNs, is that the different objectives in 12 behave on different scales, leading to instabilities in training as the gradients of the larger objective functions dominate training.

This issue is addressed by weighting each constraint term individually

$$\min_{\theta} O(\theta) + \sum_{i=0}^{m} \mu_{i_k} C_i^2(\theta). \qquad (13)$$

Besides manual tuning of the weights $\mu_{i_k}$, several schemes to dynamically balance the different terms throughout training have been proposed, such as loss-balancing via the sub-gradients ((Wang et al., 2021)), via the eigenvalues of the neural tangent kernel (Wang et al., 2022) or using a soft-attention mechanism (McClenny & Braga-Neto, 2020).

A different method for solving 11 is the augmented Lagrangian method (ALM) defined as:

$$\min_{\theta} \max_{\lambda,\mu} \mathcal{L}(\theta, \lambda, \mu) := O(\theta) + \sum_{i=0}^{m} \lambda_i C_i(\theta) + \frac{1}{2}\mu_0 \sum_{i=0}^{m} C_i^2(\theta) \,. \tag{14}$$

Using the min-max inequality or weak duality

$$\max_{\lambda,\mu} \min_{\theta} \mathcal{L}(\theta, \lambda, \mu) \leq \min_{\theta} \max_{\lambda,\mu} \mathcal{L}(\theta, \lambda, \mu) \tag{15}$$

we can solve the max-min problem instead. In each epoch $k$, a minimization over network parameters $\theta_k$ is performed using gradient descent, yielding new parameters $\theta_{k+1}$. Then, the Lagrange multipliers are updated as follows:

$$\lambda_{i_{k+1}} = \lambda_{i_k} + \mu_{0_k} C_i(\theta_{k+1}) \qquad\qquad \forall i \in 0, \dots, m. \tag{16}$$

Note that this so-called dual update of the Lagrange multipliers is simply a gradient ascent step with learning rate $\mu_{0_k}$ for each multiplier $\lambda_{i_k}$. Typically, there is also an increase of $\mu_{0_k}$ up to maximum value $\mu_{max}$ as in the penalty method. Constrained optimization with neural networks using the ALM has been shown to perform well in previous works, such as in (Son et al., 2023), (Kotary & Fioretto, 2024), (Sangalli et al., 2021), (Fioretto et al., 2021), and (Basir & Senocak, 2023).

In this classical ALM formulation, there is only a single penalty parameter $\mu_0$, which is monotonically increased during optimization. As outlined above, this is often insufficient to handle diverse constraints with different scales. Thus, we opt for the adaptive ALM proposed in (Basir & Senocak, 2023) using adaptive penalty parameters for each constraint, solving 11 as the unconstrained optimization problem:

$$\max_{\lambda} \min_{\theta} \mathcal{L}(\theta, \lambda, \mu) := o(\theta) + \sum_{i=0}^{m} \lambda_i C_i(\theta) + \frac{1}{2} \sum_{i=0}^{m} \mu_i C_i^2(\theta) \tag{17}$$

In each epoch $k$, again a minimization step over the parameters $\theta_k$ via gradient descent is performed. Then the penalty parameters $\mu_{i_k}$, which are simultaneously the learning rate of the Lagrange multipliers $\lambda_{i_k}$, are updated using RMSprop followed by the gradient ascent step for $\lambda_{i_k}$

$$\bar{\nu}_{i_{k+1}} \leftarrow \alpha\bar{\nu}_{i_k} + (1-\alpha)C_i^2(\theta_{k+1}) \tag{18}$$

$$\mu_{i_{k+1}} \leftarrow \frac{\gamma}{\sqrt{\bar{\nu}_{i_k}} + \epsilon} \tag{19}$$

$$\lambda_{i_{k+1}} \leftarrow \lambda_{i_k} + \mu_{i_k} C_i(\theta_{k+1}) \tag{20}$$

where $\bar{\nu}_i$ is the weighted moving average of the squared gradient w.r.t. $\lambda_i$, $\alpha$ is the discounting factor for old gradients, $\gamma$ is a global learning rate and $\epsilon$ is a constant added for the numerical stability of the division. This adaptive approach enables us to handle the diverse set of constraints in GINNs without the need for manual hyperparameter tuning.

Algorithm 1 shows the full algorithm used to train for $\mathcal{T}$ epochs and specifies the hyperparameters we used. The only difference to (Basir & Senocak, 2023) is that we set $\alpha = 0.90$, which is the default value of RMSprop in PyTorch, instead of $\alpha = 0.99$.

---

**Algorithm 1** Adaptive augmented Lagrangian method

---

1: **Parameters:** $\gamma = 1 \times 10^{-2}$, $\alpha = 0.90$, $\epsilon = 1 \times 10^{-8}$
2: **Input:** $\theta_0$
3: **Initialize:** $\lambda_{0,i} \leftarrow 1$, $\mu_{0,i} \leftarrow 1$, $\overline{v}_{0,i} \leftarrow 0 \; \forall i$
4: **for** $t \leftarrow 1$ to $\mathcal{T}$ **do**
5:    $\theta_t \leftarrow \mathrm{argmin}_\theta \mathcal{L}(\theta_{t-1}; \lambda_{t-1}, \mu_{t-1})$ {primal update: a gradient descent step over $\theta$}
6:    $\overline{v}_{t,i} \leftarrow \alpha \overline{v}_{t-1,i} + (1-\alpha)C_i(\theta_t)^2 \; \forall i$
7:    $\mu_{t,i} \leftarrow \frac{\gamma}{\sqrt{\overline{v}_{t,i}+\epsilon}} \; \forall i$ {penalty update}
8:    $\lambda_{t,i} \leftarrow \lambda_{t-1,i} + \mu_{t,i}C_i(\theta_t) \; \forall i$ {dual update}
9: **end for**
10: **Output:** $\theta_t$

---

### D.1. Loss plots

In Figures 11 and 12, we show the loss plots for training single and multiple shapes, respectively. As expected, the unweighted losses (middle rows in the Figures) decrease, while the Lagrange terms (bottom rows) increase over training.

## E. Topology loss

We provide additional details on our approach to the connectedness loss. We break this down in three parts: First, we define the signed distance function of a shape $\Omega$, which the neural field we train approximates. Then, we give a short rundown on computing the persistent homology (PH), in particular the PH of a neural field in a non-rectangular region. Lastly, we explain how to obtain a differentiable loss on the field from the outputs of the non-differentiable PH computation.

**Signed distance function** (SDF) $f : X \to \mathbb{R}$ of a shape $\Omega$ gives the (signed) distance from the query point $x$ to the closest boundary point:

$$f(x) = \begin{cases} d(x, \partial\Omega) & \text{if } x \in \Omega^c \text{ (if } x \text{ is outside the shape)}, \\ -d(x, \partial\Omega) & \text{if } x \in \Omega \text{ (if } x \text{ is inside the shape)}. \end{cases} \tag{21}$$

A point $x \in X$ belongs to the medial axis if its closest boundary point is not unique. The gradient of an SDF obeys the eikonal equation $\|\nabla f(x)\| = 1$ everywhere except on the medial axis, where the gradient is not defined. In INS, the SDF is approximated by a NN with parameters $\theta$: $f_\theta \approx f$.

Connectedness refers to an object $\Omega$ consisting of a single connected component. It is a ubiquitous feature enabling the propagation of mechanical forces, signals, energy, and other resources. Consequently, connectedness is an important constraint for enabling GINNs. In the context of machine learning, connectedness constraints have been multiply applied in segmentation (Wang et al., 2020; Clough et al., 2022; Hu et al., 2019), surface reconstruction (Brüel-Gabrielsson et al., 2020), and 3D shape generation with voxels (Nadimpalli et al., 2023), point clouds (Gabrielsson et al., 2020), and INSs (Mezghanni et al., 2021).

### E.1. Persistent Homology

Persistent homology (PH) is one of the primary tools that has emerged from topological data analysis to extract topological features from data. Data modalities such as point clouds, time series, graphs, and $n$-dimensional images can all be transformed into weighted cell complexes from which the homology can be computed. The homology provides global information about the underlying data and is generally robust.

**Homology** is an invariant originating from algebraic topology. A topological space $X$ is encoded as cell complexes $C_n(X)$ consisting of n-dimensional balls $B^n$ ($n = 0, 1, 2, ...$) and boundary maps $\partial_n$ from dimension $n$ to $n-1$ which satisfy $\partial_n \circ \partial_{n+1} = 0$ and $\partial_0 = 0$. The homology $H_n(X)$ is then defined as the quotient space

$$H_n(X) = \frac{\ker(\partial_n)}{\mathrm{im}(\partial_{n+1})} \tag{22}$$

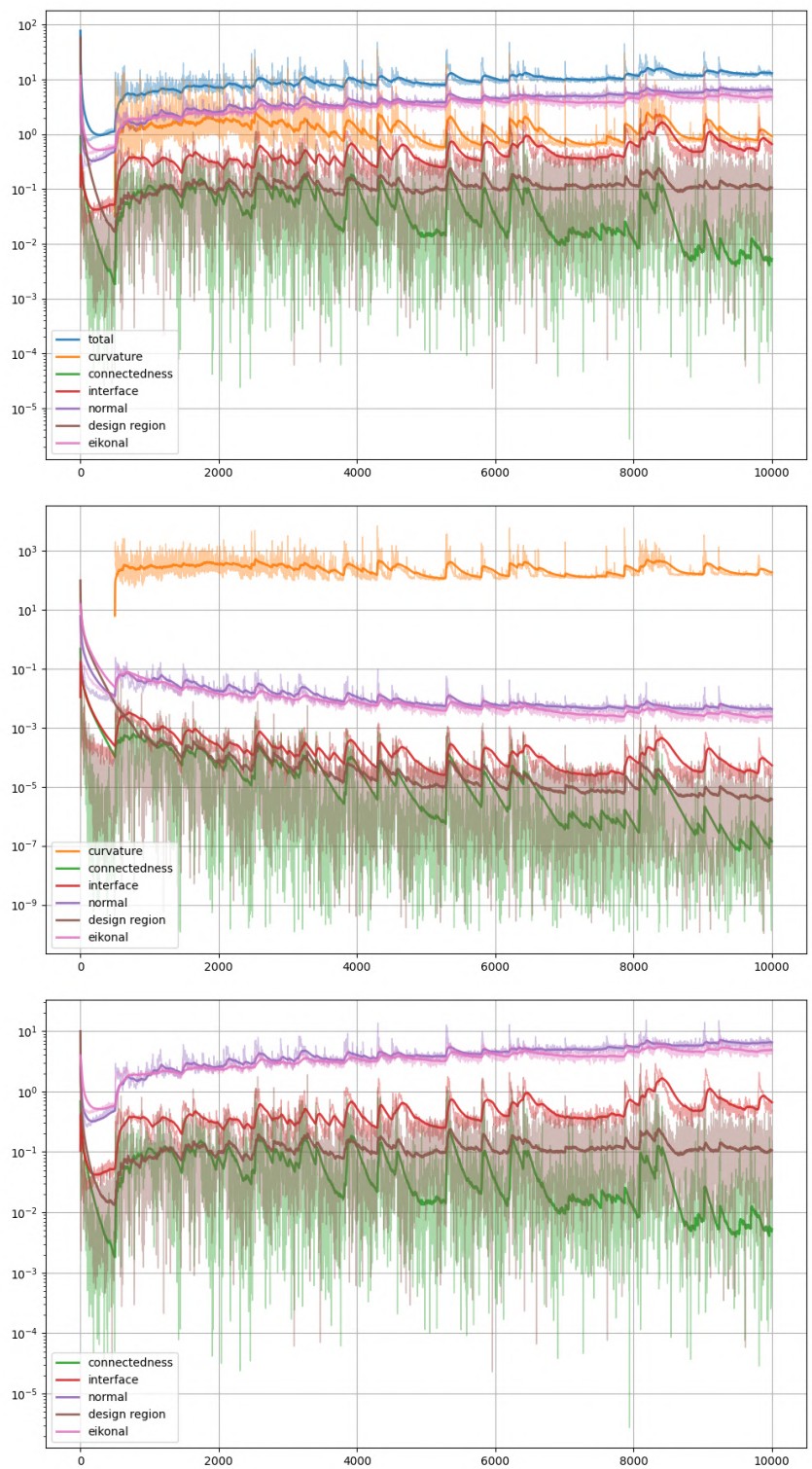

Figure 11: Loss plots for training a *single* shape of the bracket problem. The solid lines are exponential-moving averages (factor 0.99) of the noisy values in lighter colors. (a) The losses used for backpropagation. (b) The unweighted losses of each constraint. (c) The $\lambda$ values of each constraint.

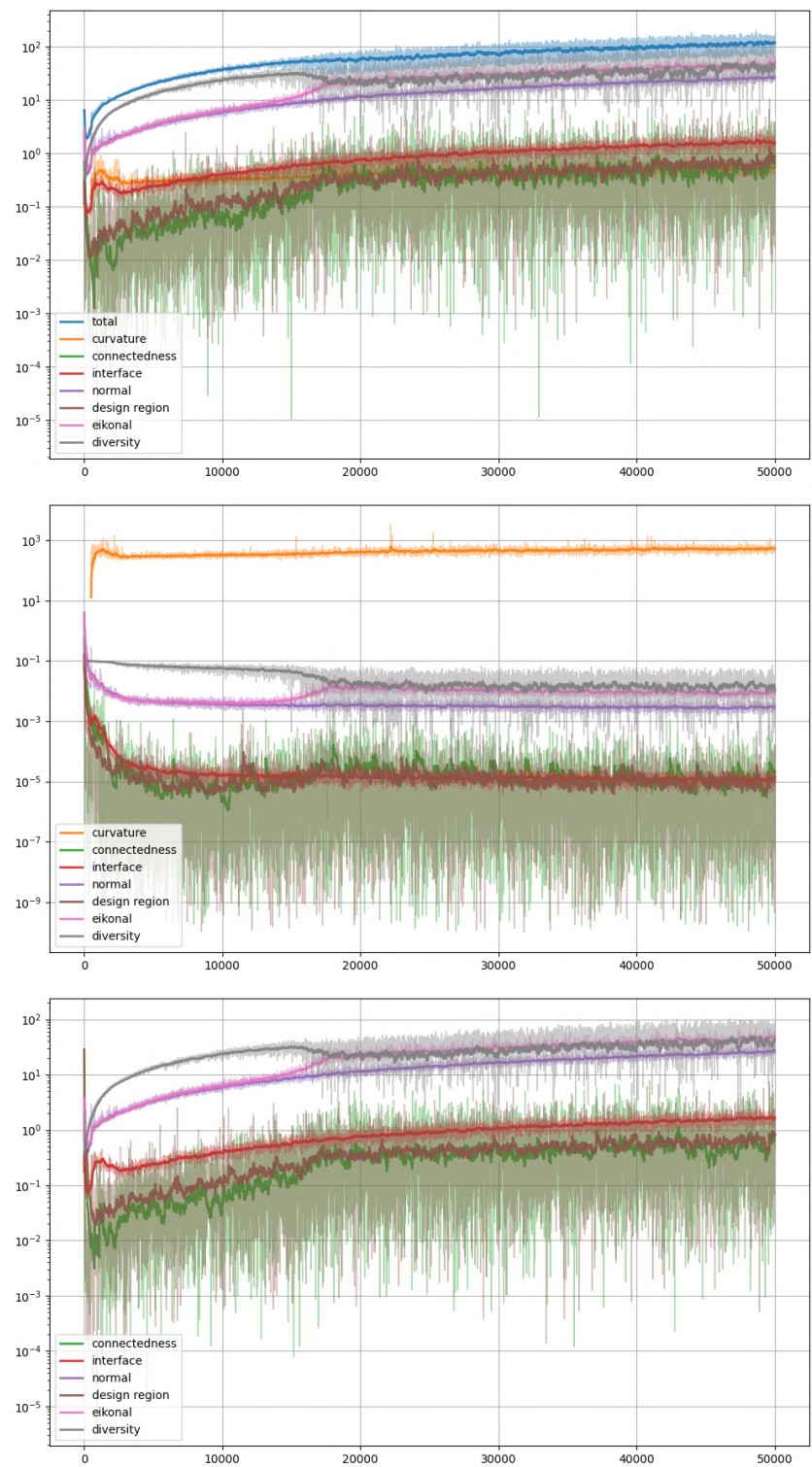

Figure 12: Loss plots for training *multiple* shapes of the bracket problem. The solid lines are exponential-moving averages (factor 0.99) of the noisy values in lighter colors. (a) The losses used for backpropagation. (b) The unweighted losses of each constraint. (c) The $\lambda$ values of each constraint.

The dimension of $H_n(X)$ counts the number of $n$-dimensional features and defines the Betti number $b_n$: for $n = 0$ the number of connected components, for $n = 1$ the number of holes, for $n = 2$ the number of voids.

**Filtrations** on the space $X$ are defined using a filter function $f : X \rightarrow \mathbb{R}$. Using a sequence of increasing parameters $\alpha_n$ with $\alpha_k < \alpha_n$ for $k < n$ we can define a sequence of nested subspaces of $X$ as sub-level sets $X_n = f^{-1}([-\infty, \alpha_n])$. We then have

$$\varnothing \subseteq X_1 \subseteq \cdots \subseteq X_N = X \ . \tag{23}$$

The homology of each of these nested complexes $C_n(X_i)$ can be computed.

**Persistent Homology** encodes how the homology of an increasing sequence of complexes changes under a given filtration. Topological features appear and vanish as the filter function sweeps over $X$. The *birth time* $b$ of a feature is defined as the value $\alpha_n$ at which the homology of $C_n(X_n)$ changes to include this feature. The *death time* $d$ of a feature is analogously defined as the value $\alpha_n$ at which it is removed from $C_n(X_n)$. The *persistence* of a feature is defined as the length of its lifetime $l = d - b$.

For each Betti number $b_n$ (for each homology class $H_n$) the information about the persistent homology of a given filtration is encoded in a persistence diagram containing the points $(b, d)$ of the birth and death pairs of all $n$-dimensional topological features (changes in the dimension of $H_n$). For a sufficiently fine filtration, the persistence diagrams contain the entire topological information about the underlying space or shape.

To compute the persistent homology of a neural field, we evaluate the network on a cubical complex on the domain of the field, i.e., a grid in $\mathbb{R}^N$. The output is simply a gray-scale image (since we are only dealing with scalar fields in this work), and the PH can be computed with existing algorithms. The current state-of-the-art algorithm for PH computation on cubical complexes is CRipser (Kaji et al., 2020).

Given a grayscale image and a filtration value $a$, the *sublevel set* at $a$ is the binary image resulting from thresholding the image for values smaller or equal to $a$. For every such binary image, which defines a weighted cubical complex with coefficients in $\mathbf{Z}/2\mathbf{Z}$, the homology can be computed. The persistence homology is then obtained by sweeping the thresholding value $a$ through $\mathbb{R}$.

In general, we are interested in computing the PH within a given design region or envelope, which is not necessarily a rectangular region. We achieve this by sampling the field in a rectangular domain containing the envelope and setting the value of points not in the envelope to $\infty$. Applying the PH computation to this altered image then correctly returns the evolution of persistence features within the envelope. The only drawback of this method is the additional computational cost of having to include the grid points outside the envelope in the PH computation, which is why the bounding domain should be chosen tightly around the envelope.

The PH computation itself does not have to be differentiable (and the CRipser library we use is not) because the cells, i.e., the grid points of the image, at which a given persistence feature is born or killed, are stored. Hence, we can simply use the network output at this grid coordinate to compute the loss and there are no issues concerning differentiability or having to re-implement the PH computation into PyTorch.

### E.2. Differentiable topology loss

To compute a differentiable loss, we use the outputs of the PH computation: For each homology class $H_n$ we obtain the points ${}^n p_i$ in the persistence diagram with the associated birth and death times ${}^n b_i$, ${}^n d_i$ and the coordinates of these births $x^{n}{}_{b_i}, y^{n}{}_{b_i}, z^{n}{}_{b_i}$ and deaths $x^{n}{}_{d_i}, y^{n}{}_{d_i}, z^{n}{}_{d_i}$.
Remark: The representatives of a homology class are not uniquely determined. The CRipser library internally chooses a representative and then outputs its coordinates. In practice, this caused no issues.

For a selected iso-level $a_0$ we select all ${}^n p_i$ for which ${}^n b_i < a_0 < {}^n d_i$ and sort them by lifetime ${}^n l_i = {}^n d_i - {}^n b_i$. Now let the index $i$ run from $1 \ldots M$ sorting the selected ${}^n p_i$. To train the network $f_\theta$ to produce a single connected component at iso-level $a_0$ the loss is given by the residuals of the deaths ${}^n d_i$ to $a_0$ for all $i = 2 \ldots M$, effectively pushing down all but the most persistent component.

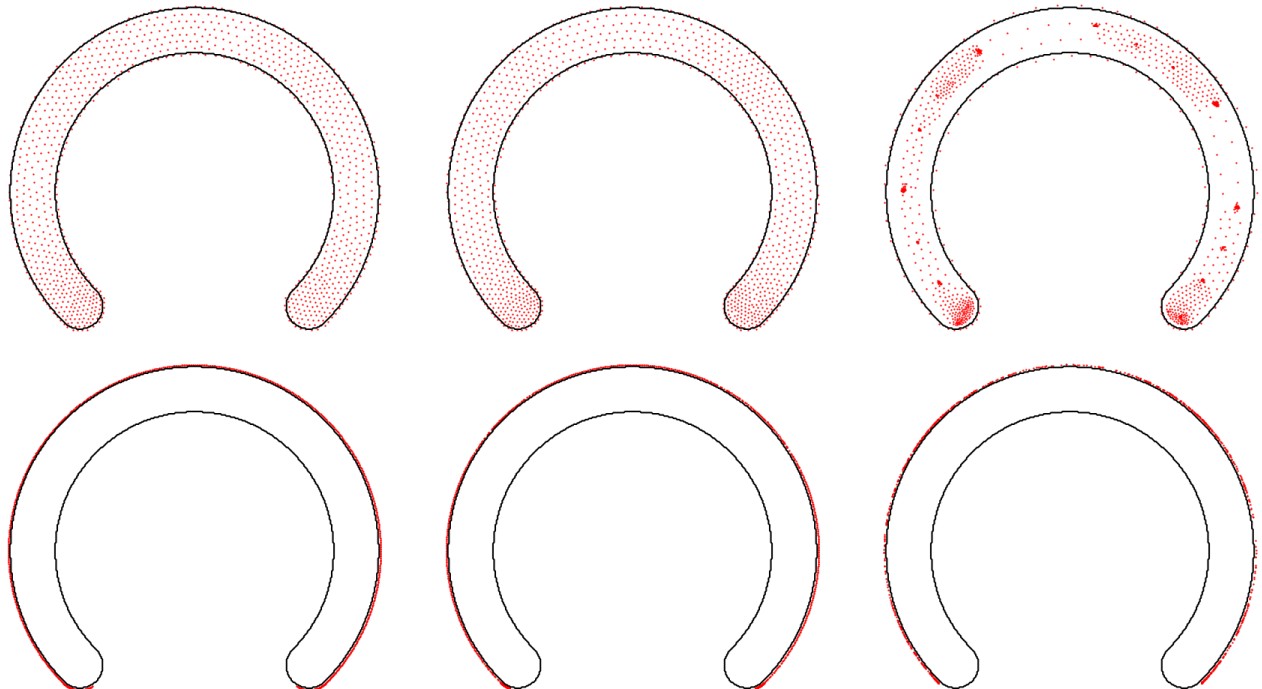

Figure 13: A visual comparison of different diversity losses in a simple 2D example ($\mathcal{F} = \mathbb{R}^2$ and the feasible set $\mathcal{K}$ is the partial annulus). Each point $f \in \mathcal{F}$ represents a candidate solution. The points are optimized to maximize the diversity within the feasible set. The top row shows the *minimal aggregation* $\delta_{\min}$ as defined in Equation 26. The bottom row shows the *total aggregation* $\delta_{\text{sum}}$ as defined in Equation 27. Each column uses a different exponent $p \in \{0.5, 1, 2\}$. For $0 \leq p \leq 1$ the minimal aggregation diversity $\delta_{\min}$ is concave meaning it favors increasing smaller distances over larger distances. This leads to a uniform coverage of the feasible set. In contrast, the $\delta_{\min}$ is convex for $p \geq 1$ as indicated by the formed clusters for $p = 2$. Meanwhile, $\delta_{\text{sum}}$ pushes the points to the boundary of the feasible set for all $p$.

$$\mathcal{L}_{cc} = \sum_{i=2}^{M} \left( a_0 - f_\theta(x_{0_{d_i}}, y_{0_{d_i}}, z_{0_{d_i}}) \right)^2 \tag{24}$$

It is immediately clear that this term is differentiable with respect to $\theta$.

More generally, to obtain a shape with a Betti number $b_n = m$ at iso-level $a_n$, the summation above runs from $i = m+1 \ldots M$. The full topology loss for an $N$-dimensional shape is then given as

$$\mathcal{L}_{topo} = \sum_{n=0}^{N-1} \sum_{i=m+1}^{M} \left( a_n - f_\theta(x_{n_{d_i}}, y_{n_{d_i}}, z_{n_{d_i}}) \right)^2 \tag{25}$$

## F. Diversity

**Concavity.** We elaborate on the aforementioned *concavity* of the diversity aggregation measure with respect to the distances. We demonstrate this in a basic experiment in Figure 13, where we consider the feasible set $\mathcal{K}$ as part of an annulus. For illustration purposes, the solution is a point in a 2D vector space $f \in \mathcal{X} \subset \mathbb{R}^2$. Consequentially, the solution set consists of $N$ such points: $S = \{f_i \in \mathcal{X}, i = 1, \ldots, N\}$. Using the usual Euclidean distance $d_2(f_i, f_j)$, we optimize the diversity of $S$

within the feasible set $\mathcal{K}$ using minimal aggregation measure

$$\delta_{\min}(S) = \left( \sum_i \left( \min_{j \neq i} d_2(f_i, f_j) \right)^p \right)^{1/p} , \tag{26}$$

as well as the total aggregation measure

$$\delta_{\text{sum}}(S) = \left( \sum_i \left( \sum_j d_2(f_i, f_j) \right)^p \right)^{1/p} . \tag{27}$$

Using different exponents $p \in \{1/2, 1, 2\}$ illustrates how $\delta_{\min}$ covers the domain uniformly for $0 \leq p \leq 1$, while clusters form for $p > 1$. The total aggregation measure always pushes the samples to the extremes of the domain.

**Distance.** We detail the derivation of our geometric distance. We can partition $\mathcal{X}$ into four parts (one, both or neither of the shape boundaries): $\partial\Omega_i \setminus \partial\Omega_j, \partial\Omega_j \setminus \partial\Omega_i, \partial\Omega_i \cap \partial\Omega_j, \mathcal{X} \setminus (\partial\Omega_i \cup \partial\Omega_j)$. Correspondingly, the integral of the $L^p$ distance can also be split into four terms. Using $f(x) = 0 \ \forall x \in \partial\Omega$ we obtain

$$
\begin{aligned}
d_2^p(f_i, f_j) &= \int_{\mathcal{X}} (f_i(x) - f_j(x))^p \, \mathrm{d}x \\
&= \int_{\partial\Omega_i \setminus \partial\Omega_j} (0 - f_j(x))^p \, \mathrm{d}x + \int_{\partial\Omega_j \setminus \partial\Omega_i} (f_i(x) - 0)^p \, \mathrm{d}x \\
&\quad + \int_{\partial\Omega_i \cap \partial\Omega_j} (0 - 0)^p \, \mathrm{d}x + \int_{\mathcal{X} \setminus (\partial\Omega_i \cup \partial\Omega_j)} (f_i(x) - f_j(x))^p \, \mathrm{d}x \\
&= \int_{\partial\Omega_i \setminus \partial\Omega_j} f_j(x)^p \, \mathrm{d}x + \int_{\partial\Omega_j \setminus \partial\Omega_i} f_i(x)^p \, \mathrm{d}x + \int_{\mathcal{X} \setminus (\partial\Omega_i \cup \partial\Omega_j)} (f_i(x) - f_j(x))^p \, \mathrm{d}x \\
&= \int_{\partial\Omega_i} f_j(x)^p \, \mathrm{d}x + \int_{\partial\Omega_j} f_i(x)^p \, \mathrm{d}x + \int_{\mathcal{X} \setminus (\partial\Omega_i \cup \partial\Omega_j)} (f_i(x) - f_j(x))^p \, \mathrm{d}x
\end{aligned}
\tag{28}
$$

## G. Geometric constraints

In Table 8, we provide a non-exhaustive list of more constraints relevant to GINNs.

| Constraint | Comment |
|---|---|
| Volume | Non-trivial to compute and differentiate for level-set function (easier for density). |
| Area | Non-trivial to compute, but easy to differentiate. |
| Minimal feature size | Non-trivial to compute, relevant to topology optimization and additive manufacturing. |
| Symmetry | Typical constraint in engineering design, suitable for encoding. |
| Tangential | Compute from normals, typical constraint in engineering design. |
| Parallel | Compute from normals, typical constraint in engineering design. |
| Planarity | Compute from normals, typical constraint in engineering design. |
| Angles | Compute from normals, relevant to additive manufacturing. |
| Curvatures | Types of curvatures, curvature variations, and derived energies. |
| Euler characteristic | Topological constraint. |

Table 8: A non-exhaustive list of geometric and topological constraints relevant to GINNs but not considered in this work.

