# OpenReview forum: "Geometry-Informed Neural Networks"
_ICML.cc/2025/Conference — ICML 2025 poster_

### Official Review · Reviewer_q1uT · 2025-03-09

**Overall Recommendation:** 3

**Summary:**

The paper proposes geometry-informed neural network (GINN) a general framework that allows for generating a diverse set of implicit shapes all satisfying a set of constraints while minimizing an objective function. The paper formulates this problem as a probabilistic generative problem with a novel diversity loss that prevents the model from mode-collapsing. It also provides a set of constraints including connectedness, smoothness, interface, and design region that can be used to enforce design constraints.

## Post Rebuttal
I appreciate the authors' rebuttal addressing most of my concerns. Therefore, I'd like to keep my original rating as it is. However, as discussed by other reviewers dCHc and 9JbR, I agree that the current experimental results do not validate the robustness of the system when optimizing for a wide range of setups. Thus, I also encourage the authors to include more settings in their experiment section for their revised version.

**Claims And Evidence:**

The paper claims to introduce GINN, a new type of neural field that is able to optimize for a given constraint while satisfying a set of constraints. The optimized network is further able to create diverse outputs all of which satisfy the constraint to some extent.

This claim is well supported by the experiments that the paper shows in the paper and the supplement.

**Essential References Not Discussed:**

N/A

**Experimental Designs Or Analyses:**

The experiment design is comprehensive with lots of qualitative and quantitative metrics provided to support the effectiveness of GINN. The ablation study in the supplement is also helpful in analyzing the effect of different losses and components of GINN. Overall I found the experiments to be complete.

**Methods And Evaluation Criteria:**

The paper proposed a general framework that is able to handle a wide range of modeling tasks. To support this claim the authors evaluated GINN on multiple tasks all of which require using GINN on different sets of data. The paper also provides an extensive ablation study which I found very helpful.

**Other Comments Or Suggestions:**

N/A

**Other Strengths And Weaknesses:**

N/A

**Questions For Authors:**

Overall I don't have major concerns regarding the paper. I found its methodology insightful and experiment results complete.

However, the experiments do seem like they are fake problems and do not have any significant practical applications. For example, in designing the engine bracket, often the design criteria cannot be straightforwardly expressed as a differentiable constraint as those presented in the paper (e.g., parameters only obtainable after physics simulation). How would the paper tackle such cases?

**Relation To Broader Scientific Literature:**

The paper proposed the first data-free generation method that is able to generate a diverse set of outputs given a set of constraints and an objective function. This can potentially open up a new interesting research direction.

**Theoretical Claims:**

I did not find discrepancies in the theoretical claims made by the paper. Although I did not go into details into the derivations presented in the supplement.

---

> ### Author Rebuttal · Authors · 2025-04-01
>
> We thank the reviewer for their positive feedback. We would merely like to answer the question concerning the extension of the method to constraints that require running a physics solver. Conceptually, this is straightforward by adopting a solver-in-the-loop [1], which we already do with the persistent-homology (PH) constraint. Loosely speaking, such a solver maps an input geometry to a differentiable scalar objective by discretizing the domain and solving an algebraic problem. This high-level procedure is the same for a PDE solver. Our framework can reuse PDE solvers from classical (differentiable) topology optimization, e.g., [2].
>
> We are currently developing the use of a PDE solver with the proposed framework in a follow-up work. Despite the conceptual similarity, there are technical differences; for example, PDE solvers are generally more expensive, require more careful discretization, and have more complicated loss landscapes, which is the reason we chose to omit this discussion in this submission.
>
>
> [1] Um, K., Brand, R., Fei, Y., Holl, P., and Thuerey, N. Solver-in-the-Loop: Learning from Differentiable Physics to Interact with Iterative PDE-Solvers. Advances in Neural Information Processing Systems, 2020.
>
> [2] Jia, Y., Wang, C., and Zhang, X. S. FEniTop: a simple FEniCSx implementation for 2D and 3D topology optimization supporting parallel computing. Structural and Multidisciplinary Optimization, August 2024.

---

### Official Review · Reviewer_dCHc · 2025-03-14

**Overall Recommendation:** 2

**Summary:**

This paper studies a novel and interesting problem of learning a generative model of shapes under certain geometric constraints. It tackles this question in a simple yet effective manner: perform constrained optimization with a diversity penalty term. Some applications are included, and some analyses are conducted to show the property of the proposed framework.

## Update after rebuttal
Thanks for the rebuttal! I agree with the authors and other reviewers that the problem and the solution are both interesting. However I keep my opinion that only two use cases are not enough to fully support the soundness of the proposed method. Given that this paper also does not have adequate theoretical contribution, I do expect extensive qualitative evaluations for this paper to meet the bar of ICML. Therefore I would keep my original evaluation.

**Claims And Evidence:**

- **Claim 1: Proposes a novel and important problem.** This claim is supported by the survey of related works (Fig 2 and Sec 2). To the best of the reviewer's knowledge, the problem setting itself is novel. The potential application of constraint-driven shape design also makes sense to me.

- **Claim 2: The proposed method is effective.**  This claim is partially supported.
  - (**Q1**) This claim is **not** sufficiently supported solely from the exisiting experiments. Although the authors propose six questions, only two of them are related to the core task (shape generation, problem 5 and problem 6). Although the experiment result looks convincing, just working on one or two problems does not show the effectiveness of the proposed method on a wide range of problems.
  - (**Q2**) There is no theoretical guarantee on the convergence of the proposed optimization process. The task itself can also be ill-defined from a theoretical perspective: how should we evaluate whether the generative model indeed recovers the desired distribution, or, what is the desired distribution from the first place?

**Essential References Not Discussed:**

- (**Q4**) Enforcing geometric constraints on neural fields and/or learning generative models for neural fields have been well-studied before. There are some missing literatures to be discussed:
  - Mehta et al., A Level Set Theory for Neural Implicit Evolution under Explicit Flows
  - Yang et al., Geometry Processing with Neural Fields
  - Schwarz et al., GRAF: Generative Radiance Fields for 3D-Aware Image Synthesis

**Experimental Designs Or Analyses:**

Yes I checked the soundness/validity of the experimental designs. As mentioned above and in Q1 and Q2, the experiment design is not sound. The experiments are so sparse that we can barely make any conclusions from it.

**(Q3)** The ablation studies are also sparse. Only Fig 8. ablates the influence of different network choice for only one task. (Fig. 3 cannot be regarded as ablation study most of the time) It's expected to include more results, such as performing ablation studies on every single task, and performing ablation on other techniques used, such as optimization methods. How is the usage of ALM influencing the effectiveness? How useful is the diversity constraint in wheel and jet engine? I noted Table 2 which serves for part of the ablations I mentioned above, however no ablations are performed for other cases, e.g., wheel.

**Methods And Evaluation Criteria:**

The method itself is straightforward and sound. However the evaluation does not make sense to me. As mentioned above, only two problems studied are related to the central problem studied (shape generation). The other four problems, although providing some insights, have no direct relationship with the actual problem (shape generation).

**Other Comments Or Suggestions:**

N/A

**Other Strengths And Weaknesses:**

Beyond the points discussed above, there are several other strengths and weaknesses as detailed below:
Other strengths:

- The paper is well-written. The writing is coherent with consistent logical flow, which presents the novel task and proposed solution effectively.
- The studied problem is interesting. This problem might be useful in several communities, especially in CAD (computer-aided design).
- The properties of the learned generative model is interesting. I find Fig. 6 particularly interesting to me, as the latent space seems to be organized and interpretable.

Other weaknesses:

- The proposed system seems to be fragile and hard-to-tune. The authors use different network architectures and different training parameters for different tasks, which indicates the complexity to tune the optimization process.

**Questions For Authors:**

If we have a limited number of ground-truth data, how would the learning framework benefit from those expert data?

**Relation To Broader Scientific Literature:**

The contribution of this paper is properly discussed in Fig. 2 and Sec. 2. The closest fields, as mentioned, are neural fields, generative modeling, and theory-informed learning.

**Theoretical Claims:**

There are no proofs.

---

> ### Author Rebuttal · Authors · 2025-04-01
>
> We thank the reviewer for the detailed and balanced review. In the following, we address the main questions.
>
> ## Q1
> The reviewer writes that while the experiments look promising, their scope is insufficient and, in particular, only two studied problems are relevant to the core research question. We believe the first four problems do support the central theme by isolating two key aspects: shape optimization with NN representations (Plateau and mirror) and training a generative model through a diversity constraint (physics and obstacle). We invite the reviewer to view these tasks as ablations of solution multiplicity and shape representations. The reviewer might also find relevant the rebuttal pdf (linked below) and the discussion with reviewer 8unU, where we provide a matrix overview of all problems and tasks and discuss their variety. However, we do agree that we can strengthen the existing problems with more ablations, which we do under Q3.
>
> ## Q2
> While the reviewer is right that our method does not come with convergence guarantees, this is the case for most adjacent methods in machine learning, physics-informed learning, and topology optimization. The convergence is a complex interplay between the model, the problem, and the optimizer, we do not foresee that such guarantees are easily obtained for the investigated non-linear differential losses on moving domains.
>
> Concerning the desired distribution, our best mental model is the Boltzmann distribution induced by the objective function over the feasible set. As described in Section 2.3, this distribution is over a function space which prohibits the direct use of Boltzmann generators (BGs) and motivates the use of an explicit diversity term that plays the role of the entropy term in BGs. Since this is an unexplored setting, we also call for further investigation under limitations and future work.
>
> ## Q3
> We have performed additional studies ablating each constraint for three tasks (obstacle, wheel, bracket) with the metrics collected in the [rebuttal pdf](https://github.com/ginn-rebuttal-icml-2025/ginn-rebuttal/blob/main/GINN_rebuttal_icml_2025.pdf). The majority of these results quantitatively confirm the obvious roles of the constraints (e.g., interface losses improve interface metrics, etc.). The diversity ablation reduces the diversity in the bracket and wheel solutions roughly five and ten-fold, respectively. Notably, diversity and smoothness losses are competing (less diversity allows for lower smoothness). The reviewer might also be interested in how these ablations impact the computational speed discussed in the response to reviewer 9JbR.
>
> We also include the ablation of the augmented Lagrangian. For the simpler problems, the constraint satisfaction is similar. For the more complex bracket problem with diversity and smoothness terms, we find that ALM finds more diverse shapes with more latent structure. This is best summarized visually ([this 3x3 plot](https://postimg.cc/SJyZfVx7) of ALM ablation can be juxtaposed with the 5x5 plot in Figure 6): while the overall structure of the shape is learnt, we see almost no interesting diversity or latent structure, and there are floaters and spurious surface features. We believe this is because ALM helps provide a stronger training signal throughout the training by adaptively increasing the weight of each loss. For some losses, the Lagrangian weight changes over two orders of magnitude.
>
> ## Q4
> We thank the reviewer for the additional references.
>
> ## Other comments
> The reviewer writes that the “system is fragile and hard-to-tune”. We must briefly comment that the wheel and bracket experiments share the exact same setup. While we chose simpler models for simpler tasks, all experiments (including the generative PINN task which is very sensitive to initialization) can be repeated with WIRE. However, in under-determined problems (those without objective, specifically, obstacle), the inductive bias of the model impacts the identified solutions, so the reviewer is right that there is a setup-dependence in this sense. Correspondingly, the low-frequency biased softplus MLP provides smooth and visually pleasing solutions, motivating such a choice. The reviewer might appreciate the [following plot](https://postimg.cc/DJhtq1nS) illustrating the role of inductive biases and diversity in connection to Q3. We are open to adding this discussion and experiments to the manuscript if the reviewer suggests this would strengthen our work.
>
> Lastly, the reviewer asks how we could incorporate ground-truth data. Conceptually, this is simple by adding a supervision loss. Using auto-decoder style training, the model should learn to organize the ground-truths in the latent space (imagine placing ground-truth shapes in Figure 6). Alternatively, the latent positions of the ground-truths can be enforced with a specific interpolation in mind (e.g., vertices of a hypercube). We believe this is a very exciting future direction.

---

> > ### Comment · Reviewer_dCHc · 2025-04-04
> >
> > Thanks for the rebuttal! I agree with the authors and other reviewers that the problem and the solution are both interesting. However I keep my opinion that only two use cases are not enough to fully support the soundness of the proposed method. Given that this paper also does not have adequate theoretical contribution, I do expect extensive qualitative evaluations for this paper to meet the bar of ICML. Therefore I would keep my original evaluation.

---

### Official Review · Reviewer_9JbR · 2025-03-15

**Overall Recommendation:** 3

**Summary:**

Paper introduces geometry-informed neural networks — a gradient-based way to optimize neural implicit function (SDF) without data based on local and global geometric constraints. Authors propose a set of practically relevant constraints (particular design region, particular interface, connectedness, smoothness, topology in form of Betti numbers) and formulated them in differentiable manner to be used in proposed optimizations.  Authors demonstrate validity of the approach via a set of qualitative and quantitative evaluations.


=====POST REBUTTAL=====

As I said in my initial review, I think this is a very promising work that might lead to a lot of interesting follow-up work as well. Authors have addressed some of my concerns in the rebuttal: time ablation with respect to different constraints, question about sharp features. I maintain my rating of 'weak accept' and I am willing to champion this paper if needed. I am not going to rise my rating because I agree with reviewer dCHc on a notion that "the proposed system seems to be fragile and hard-to-tune" but I think that this is something to be studied in follow-up work.

I strongly recommend including additional discussion into revised version's supplementary of the paper.

**Claims And Evidence:**

- Proposed method can be trained with data to satisfy design constraints to produce diverse shapes. This claim is supported by method design and experimental evaluation with different sets of constraints.
- Proposed method is applicable to a variety of problems: geometric optimization, engineering design, physics. This claim is supported via toy examples in respective domains.

Overall, I find claims and provided examples convincing (see more detailed comments in other sections).

**Essential References Not Discussed:**

Mariem Mezghanni, Malika Boulkenafed, Andre Lieutier, Maks Ovsjanikov; Proceedings of the IEEE/CVF Conference on Computer Vision and Pattern Recognition (CVPR), 2021, pp. 9330-9341

Mezghanni M, Bodrito T, Boulkenafed M, Ovsjanikov M. Physical simulation layer for accurate 3d modeling. InProceedings of the IEEE/CVF Conference on Computer Vision and Pattern Recognition 2022 (pp. 13514-13523).

Terzopoulos D, Witkin A. Physically based models with rigid and deformable components. IEEE Computer Graphics and Applications. 2002 Aug 6;8(6):41-51.

Yang L, Kim B, Zoss G, Gözcü B, Gross M, Solenthaler B. Implicit neural representation for physics-driven actuated soft bodies. ACM Transactions on Graphics (TOG). 2022 Jul 22;41(4):1-0.

Schirmer L, Novello T, da Silva V, Schardong G, Perazzo D, Lopes H, Gonçalves N, Velho L. Geometric implicit neural representations for signed distance functions. Computers & Graphics. 2024 Dec 1;125:104085.

Zhao M, Wang Y, Yu F, Zou C, Mahdavi-Amiri A. SweepNet: Unsupervised Learning Shape Abstraction via Neural Sweepers. InEuropean Conference on Computer Vision 2024 Sep 29 (pp. 302-320). Cham: Springer Nature Switzerland.

Pujol E, Chica A. Rendering piecewise approximations of SDFs through analytic intersections. Computers & Graphics. 2024 Aug 1;122:103981.

Xiang H, Jianbing S, Wang J, Ji M, Zhang C. Boundary Adaptive Physics-Informed Neural Network for the Fluid Flow Around Circular Cylinders with Complex Boundary Conditions. Available at SSRN 5110040.

**Experimental Designs Or Analyses:**

Almost all experiments in the paper are done using one set of constraints (Figure 3). It is not clear how well the method generalizes to a diverse set of constraints. This is the main reason for my rating. Paper can heavily benefit from additional examples derived from diverse sets of constraints.

**Methods And Evaluation Criteria:**

- Method evaluates qualitatively in the main paper using several toy examples;
- Additional quantitative evaluation is provided in the supplementary (Tables 2 and 3), where authors introduce a set of metrics to evaluate adherence to the constraints.
- Design region adherence is evaluated via ratio of volume/surface area outside of boundary region normalized by total volume/area outside of design region;
- Fit to interface is computed via one-sided Chamfer Distance between GT interface and interface area;
- Connectedness is evaluated on the resulting shape via connected component analysis and Betti numbers.
- Diversity is computed as variance across pairwise chamfer distances across the generated set.
- Smoothness evaluated via Monte Carlo estimation of surface strain.

Overall I find qualitative measures well designed for proposed tasks and quantitative results look good to me.

However, I think that qualitative evaluation is limited — almost all qualitative examples in the paper use one particular set of constraints (Figures 3 and 6). Paper might benefit from additional qualitative examples (and corresponding quantitative examples).

**Other Comments Or Suggestions:**

I do think that paper is somewhat poorly titled: any neural implicit function that was trained on some shapes is geometry informed in some sense. I think that the name should be something like “Constraint-based optimization of neural implicit functions” but this does not hold weight in my decision.

**Other Strengths And Weaknesses:**

+ Authors provided code
- Examples in the paper could be more diverse

**Questions For Authors:**

- Am I correct that the method is only capable of producing smooth surfaces? If yes, this should probably be clearly stated in the paper.
- How well does the method scale with respect to the number of constraints? Imagine that we add constraints the same way it is done in Figure 3. Does optimization run the same amount of time for all sets of constraints? Or convergence slows down when we increase the set of constraints?
- Do you have additional qualitative examples beyond Figure 3? Especially ones that have thin wire-like constraints (e.g. shape that contains thin spring-like shape as interface; or swiss cheese interfaces with large number of small and large holes).
- Are there any failure cases for your method that you have observed?

**Relation To Broader Scientific Literature:**

This paper can be viewed as constrained optimization of neural implicit function (SDF) via set of differentiable practically relevant constraints. The main theoretical contribution here is observation that neural implicit function can be optimized via a set of differentiable constraints instead of sampled ground truth implicit function.  The main practical contribution (based on limited set of examples) is that this optimization leads to plausible shapes.

**Theoretical Claims:**

- Method optimizes neural implicit function SDF via adaptive augmented Langrangian method to satisfy a set of constraints (examples are given in Table 1). These losses are formulated in differentiable manner in supplement.

I have checked losses and differentiable formulations and I haven’t found any issues with them.

However, I think paper should discuss couple of important theoretical considerations:
- To me, it looks like the proposed method can only produce smooth surfaces. This is important limitations because in CAD design surfaces are often non-smooth;
- Authors mention that wall-clock time for the method is 10 min per single shape. Since qualitative evaluation in the paper is limited, I assume it is the same set of constraints as depicted in Figure 3. I think paper might benefit from additional ablation that investigates the number of constraints versus optimization speed trade-off.

---

> ### Author Rebuttal · Authors · 2025-04-01
>
> We thank the reviewer for their detailed review. In the following, we will address the questions.
>
> ## Sharp features
> The method is not limited to smooth surfaces. The smoothness of the surface is inherited from the NN used to represent the shapes. The smoothness properties of MLPs are governed by the activation function. We use at least $C^2$ activations to have twice continuously differentiable surfaces, for which the Hessian and, hence, curvatures are well-defined. If this is not required, sharp features can be easily represented with, e.g., ReLU MLPs. We will also add that the log-surface-strain objective promotes sharp features that can be approximately represented with $C^2$ NNs, such as WIRE (see top row in Figure 1 or [this image](https://postimg.cc/75Pn1ScB) for a close-up).
>
> ## Impact of different constraints
> Per the reviewer’s inquiry, we have done a more comprehensive constraint ablation study collecting the time per iteration and metrics. The timings reported below are for the 3D jet engine bracket problem, which has the highest number and complexity of constraints. The other experiments show similar behaviour, so we omit these for brevity but can include them upon request.
>
> Most constraints have little effect on the time per iteration. Of the total runtime, the surface strain takes ~10% (due to the Hessian) and the PH solver ~75% (expensive multi-processed CPU task). To pre-empt a potential confusion – the runtime *increases* when ablating the eikonal constraint as it destroys the geometric regularity of the implicit function, hindering efficient surface point sampling that usually takes ~15% of total time. We use an A100-SXM GPU, and the peak memory usage for a batch of 9 shapes does not exceed 16 GB.
>
> These two losses also have a strong impact on the iterations needed. As we describe around line 430, adding the smoothness loss increases the number of iterations roughly two-fold, and adding the diversity increases it roughly five-fold. The [rebuttal pdf](https://github.com/ginn-rebuttal-icml-2025/ginn-rebuttal/blob/main/GINN_rebuttal_icml_2025.pdf) also collects the impact of these ablations on the metrics.
>
> Overall, the biggest effect is not so much the number of constraints, but rather losses that are ill-conditioned as described in “Limitations and Future Work”. This conditioning issue is also known in the PINN literature, and its mitigation is an open and active research topic.
>
> | Ablated constraint | ms/it |
> |---|:---:|
> | (None) | 260 |
> | Eikonal | 291 |
> | Interface | 264 |
> | Design region | 262 |
> | Prescribed normal | 265 |
> | Diversity | 265 |
> | Surface strain | 230 |
> | Surface strain & Diversity (no boundary point sampler) | 187 |
> | Connectedness | 94 |
>
>
> ## Additional examples
> We would first like to clarify a potential misunderstanding that “almost all qualitative examples in the paper use one particular set of constraints (Figures 3 and 6)”. Figures 1, 3, and 6 illustrate the same jet engine bracket problem for presentation continuity. However, we also solve five other problems illustrated in Figures 4 and 5 and discussed in Section 4.2. We have prepared a visual overview of the problem-constraint matrix in the [rebuttal pdf](https://github.com/ginn-rebuttal-icml-2025/ginn-rebuttal/blob/main/GINN_rebuttal_icml_2025.pdf). If this does not answer the reviewer’s question for “additional qualitative examples beyond Figure 3”, we kindly ask for clarification.
>
> ## Failure cases
> A few failure cases are discussed in the Appendix of the submission. For example, Figure 10 shows the bracket trained with softplus, SIREN, and WIRE, demonstrating the importance of the model’s inductive bias. Around line 329, we also discuss the impact of the surface sampling strategy, which should ensure that point samples are distributed uniformly and that high-curvature regions are not missed. We observed that naive sampling strategies lack these properties and lead to ridge artifacts that contain all the curvature but are missed during the optimization. We provide a basic illustration [here](https://postimg.cc/s1jLspm2). We discuss another failure mode around line 415: the emergence of latent space structure is sensitive to the diversity constraint.
>
> One could also view the ablations as failure cases. Appendix B.2 and the additional ablation study now contain more examples. We are glad to produce additional figures illustrating these failure modes if the reviewer suggests this would further strengthen the submission.

---

> > ### Comment · Reviewer_9JbR · 2025-04-07
> >
> > I am glad that authors found my review 'detailed' if not 'balanced', 'positive' or 'considerate'. I also appreciate additional effort that went into the rebuttal, especially clarification on time impact for different constraints and illustration of failures of the proposed method.
> >
> > As I said in my initial review, I think this is a very promising work that might lead to a lot of interesting follow-up work as well. So, after the rebuttal, I maintain my rating of 'weak accept' and I am willing to champion this paper if needed. I am not going to rise my rating because I agree with reviewer dCHc on a notion that "the proposed system seems to be fragile and hard-to-tune" but I think that this is something to be studied in follow-up work.
> >
> > I strongly recommend including additional discussion: smoothness concerns, constraint time analysis, additional failure cases, into revised version supplementary of the paper.

---

### Official Review · Reviewer_8unU · 2025-03-18

**Overall Recommendation:** 3

**Summary:**

The paper introduces Geometry-Informed Neural Networks (GINNs), a framework that trains shape-generative neural fields without relying on data. Instead, the method leverages user-specified design constraints (e.g. connectivity, smoothness, and topology) to drive the generation of feasible shapes. A key novelty of the paper is the explicit incorporation of diversity constraint to avoid mode collapse, which is essential when multiple solutions are desired. The framework is demonstrated on a range of problems, including real-world engineering design challenges such as wheel design and jet-engine bracket design.

**Claims And Evidence:**

Most claims made in the submission supported by clear and convincing evidence. However I have some concerns on overall applicability to real-world problems.

**Essential References Not Discussed:**

References are adequate.

**Experimental Designs Or Analyses:**

The experimental designs are valid but limited to a small range & scale of applications.

**Methods And Evaluation Criteria:**

The proposed methods make sense for the problem.

**Other Comments Or Suggestions:**

Overall the method is clean, straightforward, and could inspire further studies of PINN on the problem of 3D generation.

The related work section is extremely well-written and informative. However, the authors are encouraged to incorporate a more comprehensive and generalized experiments under a more complex setting, ideally with fair baseline comparisons. The authors could consider trim the related work a bit to fit in such experiments.

**Other Strengths And Weaknesses:**

Strengths:
- The paper tackles the important problem of shape generation in domains lacking large datasets. GINN is a new paradigm that does not requiring any training data at all. The pipeline learns neural implicit fields using only analytical objectives and constraints provided by the user. This approach draws inspiration from physics-informed neural nets, and is novel in the field of 3D generation.
- A major contribution is introducing an explicit diversity term in the optimization. By penalizing similarity among solutions, the framework yields diverse shapes that all meet the design criteria. This addresses mode collapse and is particularly valuable for design tasks where multiple viable solutions are desired,
- The method allows fine-grained control of shape properties through constraint formulation. The experiments show GINNs can enforce geometrical constrains like connectivity, smooth surfaces, and topological features such as a specified number of holes in the shape. This level of control demonstrates the flexibility of the framework to incorporate domain-specific knowledge and design intent.

Weaknesses
- *Limited Evaluation of Generality*: While GINN is demonstrated on several tasks, the set of problems is still relatively narrow and somewhat tailor-made. It’s unclear how generally the method would perform on other shape domains or more complex design scenarios beyond these settins. The paper establishes its own metrics for each constraint (which is understandable, since no standard benchmarks exist), but this makes it difficult to judge if GINN’s success extends broadly. A more convincing evaluation would include additional diverse tasks or show that the method scales to higher-complexity shapes.
- *Lack of Baseline Comparisons*: The study does not compare GINN against existing approaches, largely because “data-free shape-generative modeling is an unexplored field with no established baselines”. However, the absence of any baseline or alternative (even a simplified classical method) leaves the improvements unclear. For instance, in the engineering bracket task, the authors did not compare to classical topology optimization methods or other generative strategies, making assessment more difficult.
- *Computational Cost*: Training GINNs appears computationally expensive and potentially impractical. The paper reports that even a single shape optimization (for the jet engine bracket) requires 10k iterations (1 hour), and incorporating diversity (multiple shapes) jumps to 50k iterations (72 hours) on a single GPU. Such high time costs raise concerns about scalability for real-world design use-cases. Additionally, managing many constraints (up to seven losses) is complex; while the adaptive augmented Lagrangian helps balance them, some losses (e.g. smoothness, diversity) remain hard to optimize and could significantly slow down convergence. The paper would be stronger if it discussed or mitigated these efficiency issues, or at least compared computational load with alternative methods.

**Questions For Authors:**

See Other Strengths And Weaknesses.

**Relation To Broader Scientific Literature:**

The method is likely to be significant and inspire many downstream applications.

**Theoretical Claims:**

The theoretical claims are valid.

---

> ### Author Rebuttal · Authors · 2025-04-01
>
> We thank the reviewer for their considerate review. In the following, we address the three weaknesses.
>
> ## Task diversity
> While we agree that more experiments are almost always better, we would like to present a matrix of problems and constraints that illustrates the variety of considered settings, available in the [rebuttal pdf](https://github.com/ginn-rebuttal-icml-2025/ginn-rebuttal/blob/main/GINN_rebuttal_icml_2025.pdf), together with an additional ablation study. Not captured by this matrix are also the differences in domain dimension, shape representation, and problem symmetries (e.g., the jet engine bracket has no symmetries, making this a very general problem). Since this is an unexplored research setting, unfortunately, there does not exist a catalogue of standard problems, so defining each new problem in addition to the main research question is a considerable effort. However, we agree this is necessary to develop this research direction further and hope this is addressed in future work.
>
>
> ## Baselines
> While the reviewer and we seem to agree that a fully satisfactory comparison to a baseline is difficult to produce, we have followed the reviewer’s advice and implemented at least partially comparable baselines.
>
> ### Topology optimization
> As suggested, we consider classical TO, specifically FeniTop [1] which implements the standard SIMP method with a popular FEM solver. We define a TO problem that is as similar as possible, applying a diagonal force to the top cylindrical pin interface and allowing a 7% volume fraction in the same design region. The other interfaces are fixed in the same way. The shape compliance is minimized for 400 iterations on a 104x172x60 FEM grid (taking 190 min on a 32 core CPU to give a sense of runtime, although a fair timing comparison requires a more nuanced discussion). The produced shape is visualized [here](https://postimg.cc/1n4KD0QN).
>
> Then, we compute the surface strain (the objective we use) for this TO shape and, conversely, the compliance for a GINN shape (Section 4.2; illustrated in the penultimate column of Figure 3). Unsurprisingly, both shapes perform best at the objective they were optimized for while also satisfying the constraints up to the relevant precision. At the very least, this serves as a sanity check and confirmation of the constraint satisfaction.
>
> | Metric | Topology optimization | GINN |
> |---|---|---|
> | ↓ Connectedness (0-th Betti-number) | 1 | 1 |
> | ↓ Interface (Chamfer distance) | 0.00 | 0.00 |
> | ↓ Design region (Volume outside) | 0.00 | 0.00 |
> | ↓ Curvature | 442 | **144** |
> | ↓ Compliance | **0.99** | 0.344 |
>
> ### Human-expert dataset
> A unique aspect of GINN is the data-free shape generative aspect. Comparison to classical TO is trivial since it is inherently limited to a single solution, and its diversity would be 0. Instead, we use the simJEB [2] dataset to give some intuitive estimate on the diversity of the produced results. The dataset is due to a design challenge on a related problem described in Section 4.2. The shapes in the dataset are produced by human experts, many of whom also used topology optimization. To compute the diversity metric, we sample 196 clean shapes from the simJEB dataset and 14x14 equidistant samples from the 2d latent space generative GINN model.
>
> Even though these sets are not directly comparable as they optimize for different objectives, this comparison indicates that GINNs can produce diversity on the same and larger magnitude as a dataset that required an estimated collective effort of 14 expert human years [2].
>
> | Metric | SimJEB | GINN |
> |---|---|---|
> | ↑ Diversity | 0.099 | **0.167** |
>
> ## Computational cost
> We thank the reviewer for highlighting the runtimes, which made us realize that we have reported an outdated runtime in the submission. The up-to-date runtimes on an A100-SXM GPU are 30 min for 10k iterations of a single-shape model and 4.5h for 50k iterations for the generative model, not 72h, as mistakenly reported in the submission. This large discrepancy was due to previously logging plots at every training iteration.
>
> Since reviewer 9JbR also asks about runtimes, we invite the reviewer to read our other rebuttal, where we report a more comprehensive constraint ablation study. In brief, the two difficult losses are surface strain and diversity, both in terms of runtime and iterations needed.
>
> Respecting the character limit, we are happy to continue with a more detailed discussion upon the reviewer's request.
>
> [1] Jia, Y., Wang, C., and Zhang, X. S. FEniTop: a simple FEniCSx implementation for 2D and 3D topology optimization supporting parallel computing. Structural and Multidisciplinary Optimization, August 2024.
>
> [2] Whalen et al., SimJEB: Simulated Jet Engine Bracket Dataset, Computer Graphics Forum 2021

---

### Decision · Program_Chairs · 2025-05-01

**Decision:**

Accept (poster)

**Comment:**

This is a very promising work. It presents GINN, geometry-informed neural networks that can be trained with constraints instead of usual data. This is surprising and promising- the work should generate many follow-up efforts.

There were concerns around evaluation, ablation, and effects on other contraints. The authors' rebuttal addressed many of these concerns satisfactorily. The authors are encouraged to include the additional details (about explicit constraints in the various cases) as supplemental. Authors already agreed to provide code.

The paper received 3WAs and one WR. Given the potential of the work and the quality of results/evaluation, I am happy to recommend acceptance (as is the consensus among the reviewers). The reviews and rebuttals add a lot of value to the material included in the paper (e.g., updated timing, more ablation), and authors are encourgaed to incorporate them in the final version.